# Yeast filamentation signaling is connected to a specific substrate translocation mechanism of the Mep2 transceptor

Ana Sofia Brito[1], Benjamin Neuhäuser[2], René Wintjens[3], Anna Maria Marini[1]☯*, Mélanie Boeckstaens[1]☯*

**1** Biology of Membrane Transport Laboratory, Molecular Biology Department, Université Libre de Bruxelles, Gosselies, Belgium, **2** Institute of Crop Science, Nutritional Crop Physiology, University of Hohenheim, Stuttgart, Germany, **3** Unité Microbiologie, Chimie Bioorganique et Macromoléculaire, Département RD3, Faculté de Pharmacie, Université Libre de Bruxelles, Brussels, Belgium

☯ These authors contributed equally to this work.
* amarini@ulb.ac.be (AMM); mboeckst@ulb.ac.be (MB)

**Data Availability Statement:** All relevant data are within the manuscript and its Supporting Information files.

## Abstract

The dimorphic transition from the yeast to the filamentous form of growth allows cells to explore their environment for more suitable niches and is often crucial for the virulence of pathogenic fungi. In contrast to their Mep1/3 paralogues, fungal Mep2-type ammonium transport proteins of the conserved Mep-Amt-Rh family have been assigned an additional receptor role required to trigger the filamentation signal in response to ammonium scarcity. Here, genetic, kinetic and structure-function analyses were used to shed light on the poorly characterized signaling role of *Saccharomyces cerevisiae* Mep2. We show that Mep2 variants lacking the C-terminal tail conserve the ability to induce filamentation, revealing that signaling can proceed in the absence of exclusive binding of a putative partner to the largest cytosolic domain of the protein. Our data support that filamentation signaling requires the conformational changes accompanying substrate translocation through the pore crossing the hydrophobic core of Mep2. pHluorin reporter assays show that the transport activity of Mep2 and of non-signaling Mep1 differently affect yeast cytosolic pH *in vivo*, and that the unique pore variant Mep2$^{H194E}$, with apparent uncoupling of transport and signaling functions, acquires increased ability of acidification. Functional characterization in *Xenopus* oocytes reveals that Mep2 mediates electroneutral substrate translocation while Mep1 performs electrogenic transport. Our findings highlight that the Mep2-dependent filamentation induction is connected to its specific transport mechanism, suggesting a role of pH in signal mediation. Finally, we show that the signaling process is conserved for the Mep2 protein from the human pathogen *Candida albicans*.

## Author summary

Fungal Mep2-type ammonium transport proteins of the conserved Mep-Amt-Rh family that includes human Rhesus factors are specifically required to allow filamentation in

**Funding:** ASB is a Research Fellow of the F.R.S.-FNRS, RW is a research associate of the F.R.S.-FNRS, AMM is a senior research associate of the F.R.S.-FNRS and a WELBIO investigator, and MB is a scientific research worker supported by WELBIO. AMM received support for this work from F.R.S.-FNRS (CDR J017617F, PDR T011515F, PDR 33658167), the Fédération Wallonie-Bruxelles (Action de Recherche Concertée), WELBIO, Université Libre de Bruxelles (FER), the Brachet Funds, and the "Alice et David Van Buuren" foundation. The funders had no role in study design, data collection and analysis, decision to publish, or preparation of the manuscript.

**Competing interests:** The authors have declared that no competing interests exist.

response to ammonium limitation. These proteins were therefore assigned a receptor role while the underlying mechanism of signal transduction remains poorly understood. The "transceptor" property has subsequently been proposed to concern transporters of all kind of micro- and macro- nutrients in eukaryotes, from fungi to human. However, bringing the firm demonstration of their existence remains challenging as variants with full uncoupling of transport and receptor functions are difficult to obtain. Our data question the involvement of the C-terminal extremity of *Saccharomyces cerevisiae* Mep2 in the signal mediation leading to filamentation. If signaling partners exist, they should also bind to cytosolic loops and/or membrane-embedded domains. The capacity of Mep2 to enable filamentation is closely intertwined to the mechanism of substrate translocation through the pore of the hydrophobic core of the protein. In *Xenopus* oocytes, the transport activity of non-signaling Mep1 is electrogenic while it is electroneutral for Mep2, the latter likely translocating the weak base $NH_3$, but not the proton released after $NH_4^+$ recognition and deprotonation. We propose that given consequences of a Mep2-specific transport process, such as an intracellular pH modification, could be the underlying cause of the filamentation signal ensured by Mep2-type proteins.

## Introduction

All cells need to cope with their environment, respond to nutritional, physical and mechanical constraints, and in some cases migrate towards more suitable conditions. Invading cancer cells change their shape, producing morphological asymmetry, and execute locally controlled extracellular proteolysis facilitating invasion and leading to formation of metastases ultimately developing in different environments [1]. Fungi are also able to invade their milieu by producing hyphae and extracellular hydrolases. In pathogenic fungi, the switch from budding to filamentous growth is a dimorphic transition often related to the virulence [2, 3]. Fungal infections represent a significant cause of mortality in human, and of huge damage to crop yield worldwide. Nitrogen scarcity is one among the signals triggering the filamentation process [4]. In analogy to the human pathogen *Candida albicans*, diploid cells of baker yeast *Saccharomyces cerevisiae* respond to nitrogen limitation by switching to an unipolar mode of budding thereby producing pseudohyphae that invade the agar medium, likely to scavenge the environment for nutrients [5]. The dimorphic switch for instance occurs when limiting ammonium levels constitute the sole nitrogen supply.

Ammonium is a major nitrogen source for microorganisms and plants whilst it is mostly documented for its role in pH homeostasis and for the toxicity of its accumulation in animals [6–8]. Recent data further reveal a role of ammonium in cancer cell proliferation [9, 10]. *S. cerevisiae* possesses three members of the conserved Mep-Amt-Rh protein family mediating the transmembrane transport of ammonium, hereafter referring to the sum of $NH_4^+$ and $NH_3$ molecular entities [11–14]. The proteins of this family, comprising the human Rhesus factors, display 11 or 12 transmembrane (TM) spans and a cytosolic C-terminal domain (CTD), and adopt a trimeric fold [15–19]. Of the three yeast *MEP* genes, only the specific deletion of *MEP2* impairs filamentation, growth being ensured by the ammonium transport activity of the remaining Mep1 and Mep3 paralogues [20]. The single expression of *MEP2*, in the absence of *MEP1* and *MEP3*, enables both ammonium transport for growth and filamentation induction, while the single expression of *MEP1* or *MEP3* only allows ammonium transport for growth. *S. cerevisiae* ammonium transport protein Mep2, and later on several other fungal Mep2-type

proteins, were thereby assigned an additional sensor role that is required for filamentous development in response to ammonium limitation for instance [20–25].

Several evolutionarily conserved signaling pathways regulate filamentous growth in yeast, including the STE mitogen activated protein kinase (MAPK), rat sarcoma/protein kinase A (RAS/PKA), sucrose nonfermentable (SNF), and target of rapamycin (TOR) pathways [4]. The filamentation defect of Mep2-lacking cells is suppressed by exogenous cAMP addition, by expression of a dominant active allele of Gpa2 or a Ras2 variant, all intermediates of the PKA pathway, suggesting that the transport protein operates upstream of the latter [20]. However, the Mep2-dependent signaling mechanism remains unclear. There is a general lack of information regarding the existence and the role of potential signaling Mep2 partners [26, 27]. In *Ustilago maydis*, a physical interaction is reported between the small G protein Rho1 and the Mep2-type Ump2 protein, but also with the non-signaling ammonium transport protein Ump1 [26]. Of note, no constitutive variant of Mep2 proteins able to signal in the total absence of transport has been isolated. Some studies indicate that substrate transport and filamentation induction efficiencies are intimately linked while others have reported several mutations in the conducting pore or in the CTD of Mep2-type proteins that result in the loss of filamentation induction while largely conserving substrate transport capacity, indicating apparent uncoupling of transport and signaling functions [21, 22, 25, 27–31]. This concerns specific mutations in a pair of histidine residues widely conserved in the pore of Mep-Amt-Rh transport proteins except for the fungal non-signaling Mep1-like proteins displaying a glutamate instead of the first histidine of the dyad [29]. We proposed that a difference in the transport mechanism of Mep1- and Mep2-type proteins could underlie the signaling capacity of the latter.

The exact nature of the substrate(s) translocated by Mep-Amt-Rh proteins and the employed molecular mechanism are still a matter of contention. Available experimental data suggest that, following initial recognition, $NH_4^+$ undergoes deprotonation and $NH_3$ then permeates through the hydrophobic pore while the fate of the released proton remains unclear [18, 32–38]. A difference between Mep1- and Mep2-type proteins has already been depicted and notably concerns the molecular mechanisms enabling the activity tuning by the TORC1 effector kinase Npr1 [39–41]. The kinase mediates the S457-phosphorylation and thereby the silencing of an autoinhibitory CTD in the case of Mep2 and controls the phosphorylation of an intermediate inhibitory partner, Amu1/Par32, in the case of Mep1 as well as its close paralogue Mep3 [39–41].

We show here that filamentation is intimately related to Mep2 transport activity. Mep2 variants lacking the CTD but containing specific pore mutations enabling transport can still allow filamentation induction. Substrate translocation via Mep1 or Mep2 has a different impact on cytosolic pH *in vivo* and a Mep2H194E pore variant shows increased competence to acidify the submembrane pH while losing the signaling capability. The equivalent H188E pore mutation in *C. albicans* Mep2 also enables uncoupling of transport and signaling functions upon expression in baker yeast. Furthermore, electrophysiology and ammonium uptake experiments in *Xenopus* oocytes show that non-signaling Mep1 performs electrogenic substrate transport while Mep2 mediates electroneutral substrate translocation. The Mep2-specific transport process, or an aftereffect of it such as a pH modification, could play a key role in the filamentation signaling ensured by Mep2-type proteins.

## Results

### Transport activity of Mep2 CTD variants correlates with filamentation efficiency

Fungal Mep proteins of the Mep-Amt-Rh family possess 11 TM domains with a periplasmic N-terminal domain and a cytoplasmic CTD (Fig 1A) [15, 19]. The transport activity of the

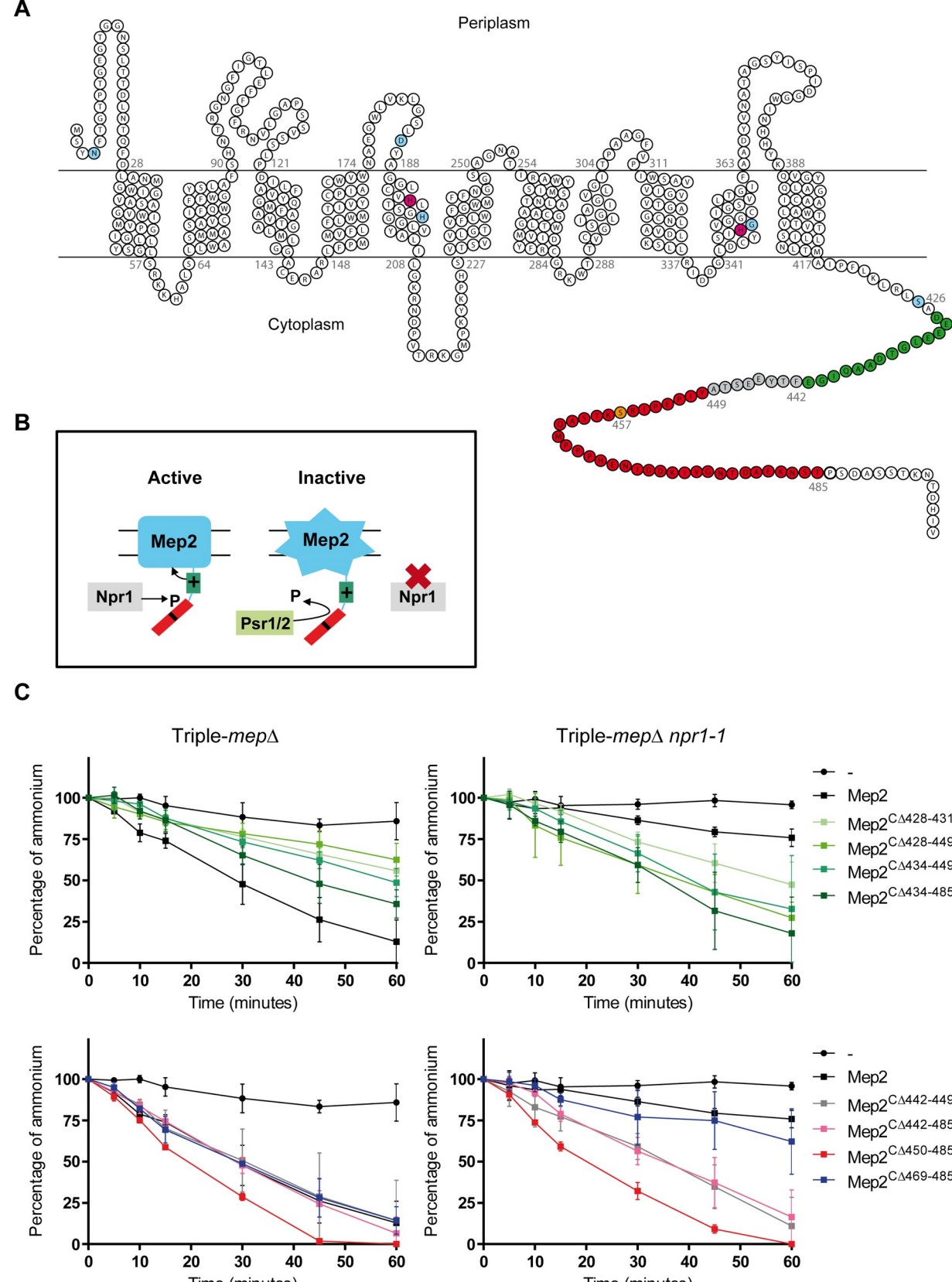

**Fig 1. The ammonium transport protein Mep2 is regulated by its C-terminal extremity and the Npr1 kinase.** (A) Topology model of the Mep2 protein with the location of the mutated residues and amino-acid deletions used in this study. In blue: the N4, D186, H199, G349 and S426; in magenta: the conserved histidine-twin H194 and H348; in orange: the phosphorylation site S457. The enhancer, linker and

autoinhibitory domains are labelled in green, grey and red, respectively. (B) Model of Mep2 regulation by the TORC1 effector kinase Npr1, modified from [39]. The transport activity of the hydrophobic core of yeast Mep2 is fine-tuned by a mechanism involving the modulation of the spatial organization of different regions in the CTD. Left panel: When the Npr1 kinase is active, it enables S457 phosphorylation and silencing of the C-terminal autoinhibitory domain (amino-acids 450–485) of Mep2. The enhancer C-terminal domain, limited to residues 428–441, is free to activate the transport protein. Right panel: When Npr1 is inactive, Mep2 is rapidly dephosphorylated by the Psr1/2 phosphatases. The non-phosphorylated autoinhibitory domain prevents the enhancer domain to activate the Mep2 protein. (C) Homozygous diploid triple-*mepΔ* (ZAM38) and triple-*mepΔ npr1-1* (ZMB058) cells were transformed with the pFL38 empty plasmid (-) or with YCpMep2, YCpMep2$^{CΔ428–431}$, YCpMep2$^{CΔ428–449}$, YCpMep2$^{CΔ434–449}$, YCpMep2$^{CΔ434–485}$, YCpMep2$^{CΔ442–449}$, YCpMep2$^{CΔ442–485}$, YCpMep2$^{CΔ450–485}$ or YCpMep2$^{CΔ469–485}$. Ammonium removal rates of cells growing in SHPD (0.1% proline) liquid medium. At time 0, 500 μM ammonium was added and its removal from the medium was monitored for 1 h. Ammonium remaining in the medium is expressed as percentage of the initial concentration. Averages and standard deviations are reported (n = 3).

hydrophobic core of yeast Mep2 is fine-tuned by a mechanism involving the modulation of the spatial organization of different regions in the CTD (Fig 1B) [39]. An enhancer domain, limited to residues 428–441, upregulates substrate translocation via the Mep2 hydrophobic core, while an autoinhibitory domain, comprised within the 450–485 region and including the Npr1-target serine S457, counteracts the action of the enhancer domain. In between, a linker domain, limited to residues 442–449, appears required for optimal Mep2 activity when the kinase is present but dispensable when the kinase integrity is altered (Fig 1).

The CTD of Mep2 could serve as a platform for the binding of signaling partners. To evaluate the role of the CTD in the receptor function of Mep2, triple-*mepΔ* diploid cells, lacking all three *MEP* genes, but expressing Mep2 variants bearing domain deletions were assayed for filamentation efficiency by dropping high-density cell suspensions onto a specific SLAD medium containing a limiting ammonium concentration (100 μM) and following the appearance of protruding filaments. A similar SHAD medium containing a higher ammonium concentration (1 mM), conditions in which filamentation is repressed, was used to check for potential constitutive induction of filamentation. In parallel, the transport functionality of Mep2 variants was evaluated by two ways. We followed the growth development of single colonies from low-density suspensions of cells streaked on SLAD to test the capacity of cells to form colonies in the condition of filamentation induction and on SHAD. We also measured the ammonium removal ability of cells exponentially growing in a similar liquid medium, SHPD, containing a non-limiting concentration of proline (0.1%) as sole nitrogen source. Although ammonium removal in SHPD can't be directly compared to growth tests on limiting SLAD, the assay gives an indication on the functionality of Mep2 variants in optimal conditions. The results of the distinct assays are recapitulated in S1 Table. As expected, growth of cells expressing Mep2 variants mutated in the enhancer domain (Mep2$^{CΔ428–431}$, Mep2$^{CΔ428–449}$, Mep2$^{CΔ434–449}$) was affected on SLAD and SHAD (Fig 2A). However, partial complementation of the growth defect was observable after prolonged incubation (7 days). All three variants were detectable by western blot although they appeared slightly less abundant than native Mep2 (Fig 2C). Of note, compared to the doublet band of native Mep2, the variants deleted in the positive domain were detected as one main band, the upper band being only barely detectable for the Mep2$^{CΔ428–431}$ variant, suggesting that the deleted region is structurally important to allow efficient phosphorylation of the autoinhibitory domain. These variants still enabled ammonium removal in SHPD though at a reduced rate compared to native Mep2 (Fig 1C). They did however not promote filamentous growth (Fig 2B). The absence of filamentation was also observed when the variants were expressed in *mep2Δ* cells in which growth is supported by ammonium uptake via endogenous Mep1 and Mep3 ammonium transport proteins (S1 Fig). Therefore, the enhancer domain appears important for both transport and signaling functions of Mep2 and a reduced transport function seems correlated with a reduced filamentation efficiency.

Cells expressing the Mep2$^{CΔ450–485}$ variant lacking the autoinhibitory domain grew comparably to those expressing native Mep2 on SLAD and SHAD but showed increased filamentation

and ammonium removal rate (Figs 1C, 2A and 2B) [39]. Hence, this Mep2 variant displaying an increased transport function enables improved filamentation. Mep2$^{C\Delta469-485}$ behaved as native Mep2 (Figs 1C, 2A and 2B). Deletion of the linker domain (Mep2$^{C\Delta442-449}$) of Mep2 had no major impact on the cells filamentation efficiency and on ammonium removal rate (Figs 1C and 2B). None of these Mep2 variants allowed filamentation in ammonium-repleted conditions (SHAD) (Fig 2B), indicating that none acquired the ability to allow constitutive filamentation.

Together, these data support a correlation between ammonium transport and signaling efficiencies of Mep2.

In several cases, we observed that a given Mep2 variant bypassing the Npr1 requirement is even more active in the absence of Npr1 than in its presence [25, 39]. For instance, this also concerns Mep2$^{C\Delta450-485}$ and the phosphomimetic Mep2$^{S457D}$ variant, suggesting that the cellular context of Npr1-lacking cells also influences the Mep2 activity in a manner independent of the autoinhibitory domain. We therefore analyzed the impact of the Npr1 absence on the ability of Mep2 CTD variants to allow filamentation.

Several variants that were functional in triple-*mepΔ* cells, namely Mep2$^{C\Delta428-449}$, Mep2$^{C\Delta434-449}$ and Mep2$^{C\Delta434-485}$, including certain ones in the enhancer domain, showed an even better complementation efficiency in a triple-*mepΔ npr1-1* background (Fig 2A). This correlated with an improved ammonium removal rate in SHPD and also with a capacity to induce filamentation on SLAD in Npr1-lacking cells (Figs 1C and 2B). These variants are thus able to adopt a conformation allowing more significant transport and filamentation in Npr1-lacking cells. The Mep2$^{C\Delta428-431}$ variant in the enhancer domain, only produced few filamentation tips and was also more affected in its ability to complement the growth defect of triple-*mepΔ npr1-1* on SLAD and displayed the slowest ammonium removal rate compared to the variants showing activity in Npr1-lacking cells (Figs 1C, 2A and 2B).

The Mep2 variants bearing truncations in the enhancer domain and expressed in cells lacking Npr1 were all detectable by western blot (Fig 2C). The intensity of the signal was stronger compared to the signal observed in the presence of Npr1, suggesting that an increased abundance of these Mep2 variants could also contribute to the better ammonium transport rate and filamentation efficiency observed in the absence of the kinase.

Variants in the linker and/or autoinhibitory domain, Mep2$^{C\Delta442-449}$, Mep2$^{C\Delta442-485}$ and Mep2$^{C\Delta450-485}$, allowed hyperfilamentation on SLAD (Fig 2B) without showing increased ammonium removal rate in SHPD in the absence compared to the presence of Npr1 (Fig 1C). However, in Npr1-lacking cells *per se*, these three variants allowed a higher ammonium removal rate in SHPD compared to Mep2$^{C\Delta428-449}$, Mep2$^{C\Delta434-449}$ and Mep2$^{C\Delta434-485}$ (Fig 1C), and were able to provide a growth advantage visible on SHAD at day 3 for instance (Fig 2A). Moreover, we previously showed that Mep2$^{C\Delta442-449}$, Mep2$^{C\Delta442-485}$, Mep2$^{C\Delta450-485}$ variants display a higher initial rate of $^{14}$C-methylammonium uptake when Npr1 is absent [39]. Few filaments were even detected on SHAD in Npr1-lacking cells expressing some variants (Mep2$^{C\Delta428-449}$, Mep2$^{C\Delta434-485}$ and Mep2$^{C\Delta442-449}$). The Mep2$^{C\Delta469-485}$ remained largely sensitive to the Npr1 absence, and unable to allow filamentation (Fig 2A and 2B).

Overall, most of these data are indicative of a correlation between transport and filamentation efficiencies.

## The S457D phosphomimetic mutation does not lock Mep2 in a conformation constitutive for signaling

The C-terminal phosphomimetic S457D mutation silences the autoinhibitory domain of Mep2, conferring transport activity even in the absence of the Npr1 kinase, and suggesting that an active conformation of the protein is stabilized [39]. Structural data support a modified

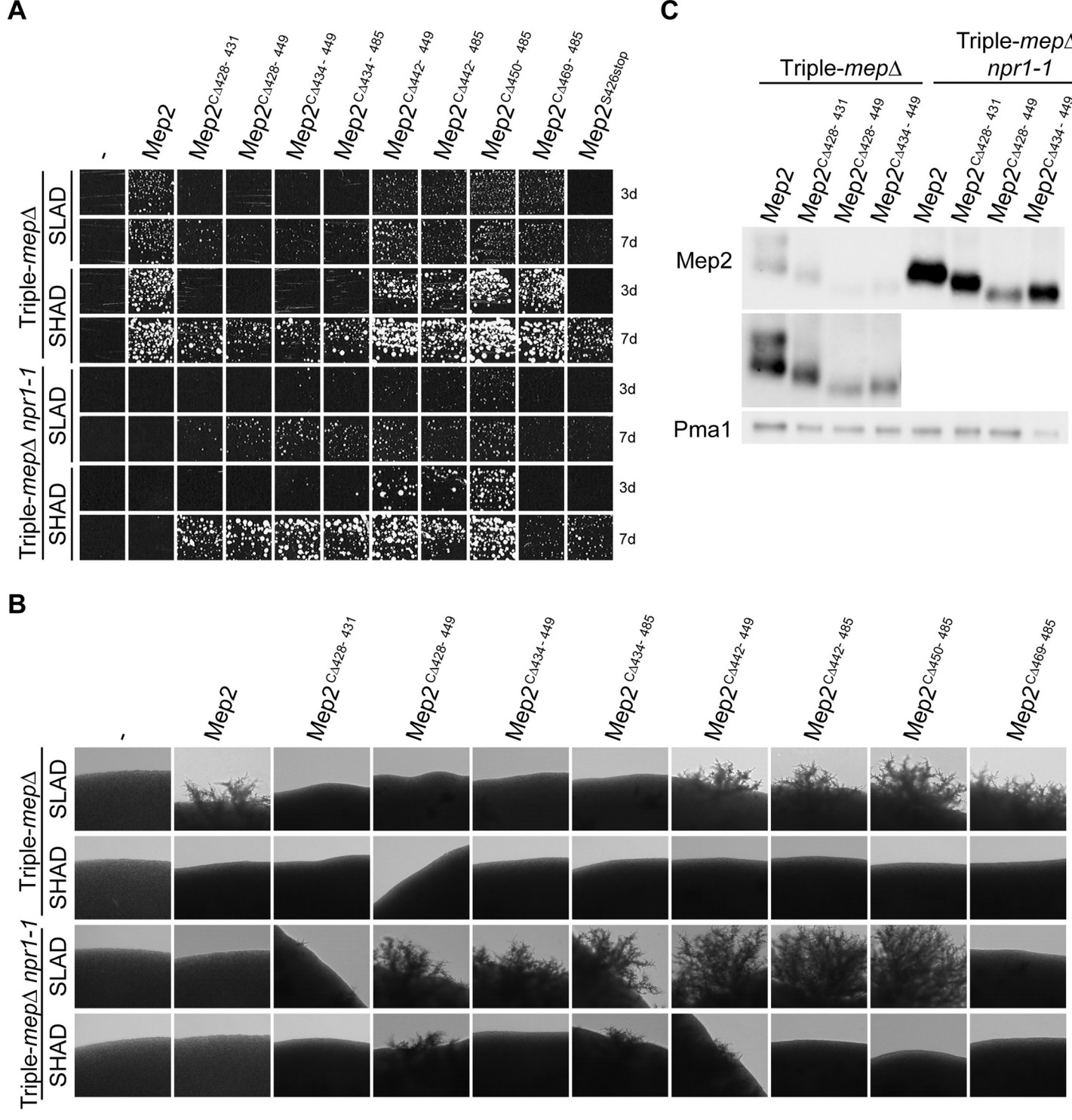

**Fig 2. Transport activity of Mep2 CTD variants correlates with pseudohyphal growth efficiency.** Homozygous diploid triple-*mepΔ* (ZAM38) and triple-*mepΔ npr1-1* (ZMB058) cells were transformed with the pFL38 empty plasmid (-) or with YCpMep2, YCpMep2$^{S426stop}$, YCpMep2$^{CΔ428–431}$, YCpMep2$^{CΔ428–449}$, YCpMep2$^{CΔ434–449}$, YCpMep2$^{CΔ434–485}$, YCpMep2$^{CΔ442–449}$, YCpMep2$^{CΔ442–485}$, YCpMep2$^{CΔ450–485}$ or YCpMep2$^{CΔ469–485}$. (A) Growth tests of low-density cell suspensions streaked on SLAD (100 μM ammonium) and SHAD (1 mM ammonium) media at day 3 (3d) and 7 (7d), at 29˚C. (B) Pseudohyphal growth tests of high-density cell suspensions dropped on SLAD and SHAD media at day 7. (C) Immunodetection of Mep2 from membrane-enriched cell extracts treated with N-glycosidase F. Cells were grown in buffered minimal medium containing 0.1% proline. Pma1 was detected as a loading control.

conformation of the CTD of a *Candida albicans* Mep2 variant with analogous phosphomimetic environment [15]. We took advantage of the S457D mutation to test whether a Mep2 protein locked in a putative active fold allows filamentation even in the absence of ammonium limitation. We further combined the S457D mutation with the D186N mutation proposed to disrupt substrate recognition [31]. In contrast to native Mep2 and Mep2$^{S457D}$, but similarly to Mep2$^{D186N}$, the Mep2$^{D186N, S457D}$ variant was not able to complement the growth defect of *triple-mepΔ* cells on SLAD or SHAD, indicating that the D186N mutation still impairs ammonium transport when combined to S457D (Fig 3A). Mep2$^{D186N, S457D}$ variant did not allow filamentation in triple-*mepΔ* cells on SLAD, as expected from the growth defect. The absence of filamentation was also observed when the Mep2$^{D186N, S457D}$ variant was expressed in a simple *mep2Δ* background in which ammonium uptake required for growth is ensured by endogenous Mep1 and Mep3 ammonium transport proteins (Fig 3B). Similarly to native Mep2, the Mep2$^{S457D}$ variant allowed filamentation on SLAD but not on SHAD, indicating the absence of constitutive filamentation under ammonium sufficiency (Fig 3B). These data also indicate that Mep2$^{S457D}$, which likely adopts preferentially a transport-active conformation, is still able to allow filamentation. We verified that D186N and S457D single and double Mep2 variants were all stably produced (Fig 3C). These variants were immunodetected at similar global levels, Mep2$^{D186N, S457D}$ protein behaved as Mep2$^{S457D}$, being detected as one major form, while Mep2 and Mep2$^{D186N}$ were detected as a doublet band, consistent with former observations (Fig 3C) [31, 39].

These results reveal that the S457D phosphomimetic mutation does not lock Mep2 in a conformation able to constitutively induce the signal leading to filamentation. They further indicate that the absence of filamentation is correlated with the absence of substrate translocation via Mep2.

We further assessed a potential role in Mep2-dependent filamentation of the plasma-membrane Psr1 and Psr2 redundant phosphatases required for the dephosphorylation of S457 [39]. Deletion of the Psr1 and Psr2 phosphatases did not alter the ability of native Mep2 or Mep2$^{S457D}$ variant to complement the growth defect of triple-*mepΔ* cells nor to allow filamentation on SLAD (Fig 3D and 3E). Furthermore, filamentation was still repressed in cells lacking the Psr phosphatases in ammonium sufficiency conditions on SHAD, indicating that stabilizing the phosphorylation of S457 is not sufficient to allow constitutive filamentation induction via Mep2.

Besides their function in Mep2 activity regulation, the Psr1 and Psr2 phosphatases do not play a role in filamentation induction.

## The Mep2 C-terminal tail can be dispensable for filamentation induction

The Mep2 hydrophobic core likely possesses a basal transport activity while the CTD plays a significant role in tuning the transport function of Mep2 [39]. The Mep2$^{S426stop}$ variant, bearing a major truncation of the CTD, poorly complemented the growth defect of triple-*mepΔ* cells on SLAD and SHAD media and is thus unable to induce filamentation on SLAD medium (Fig 4A and 4B). The absence of filamentation was also observed when the variant was expressed in a simple *mep2Δ* background (S1 Fig).

We reasoned that one way to evaluate whether the CTD of Mep2 is required for filamentation induction would be to consider the signaling properties of a variant lacking the full CTD but able to transport sufficient substrate to ensure growth. For this purpose, we characterized a spontaneous suppressor isolated for its ability to confer activity to the Mep2$^{S426stop}$ variant thereby enabling triple-*mepΔ* cells to grow on minimal medium containing 1 mM ammonium. The identified suppressor mutation, H199Y, lies in the 5$^{th}$ TM domain, at the packing interface

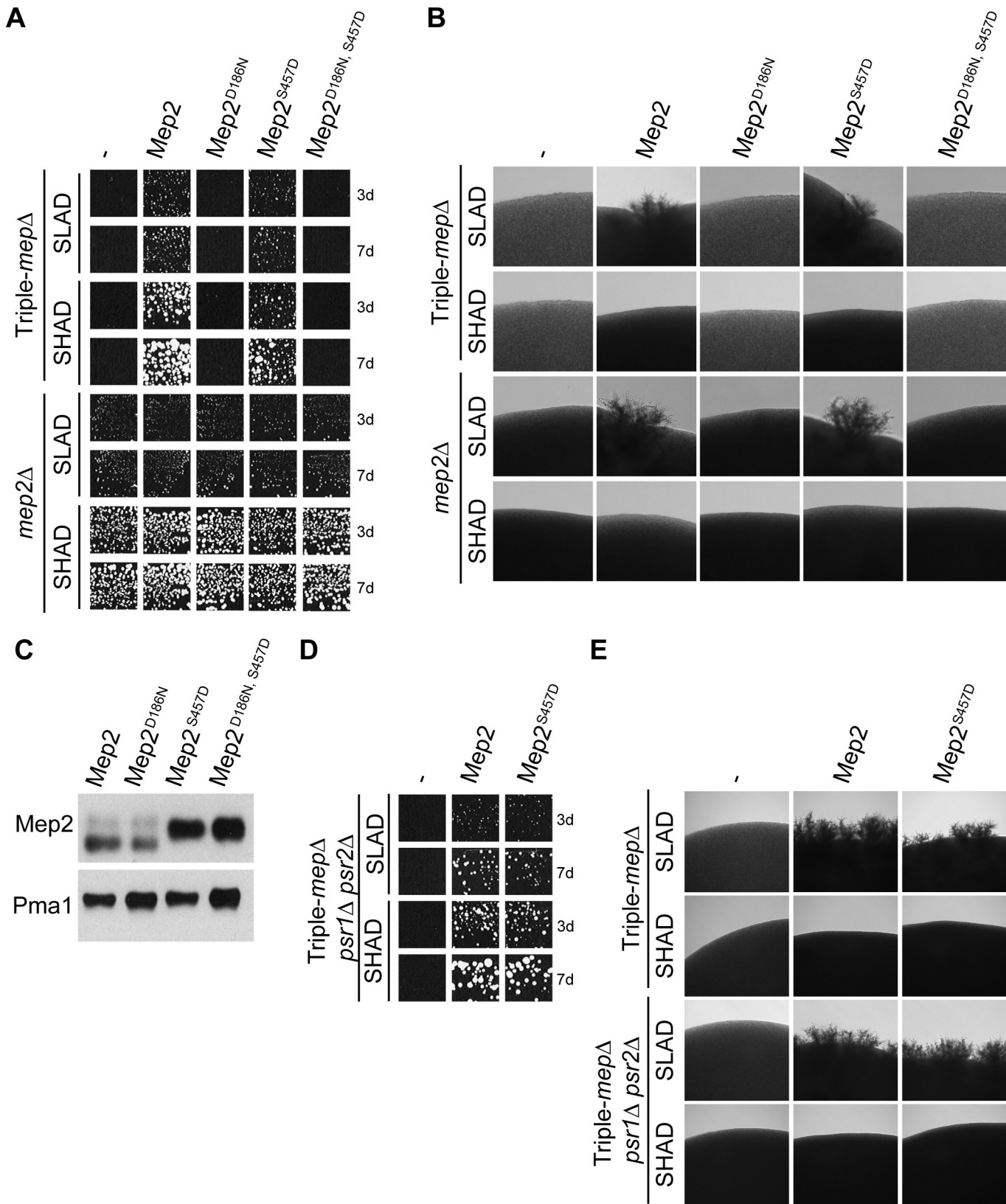

**Fig 3. The S457D phosphomimetic mutation does not lock Mep2 in a conformation allowing constitutive filamentation.** (A-B) Homozygous diploid triple-*mepΔ* (ZAM38) and *mep2Δ* (ZAB2) cells were transformed with the pFL38 empty plasmid (-) or with YCpMep2, YCpMep2$^{D186N}$, YCpMep2$^{S457D}$ or

YCpMep2$^{D186N, S457D}$. (A) Growth tests of low-density cell suspensions on SLAD and SHAD media at day 3 (3d) and 7 (7d), at 29°C. (B) Pseudohyphal growth tests of high-density cell suspensions dropped on SLAD and SHAD media at day 7. (C) Immunodetection of Mep2 from membrane-enriched extracts, treated with N-glycosidase F, of haploid triple-*mepΔ* cells (31019b) transformed with YCpMep2, YCpMep2$^{D186N}$, YCpMep2$^{S457D}$ or YCpMep2$^{D186N, S457D}$ plasmids and grown in buffered minimal medium containing 0.1% proline. Pma1 was detected as a loading control. (D-E) Homozygous diploid triple-*mepΔ* (ZAM38) and triple-*mepΔ psr1Δ psr2Δ* (ZAB1) cells transformed with the pFL38 empty plasmid (-) or with YCpMep2 or YCpMep2$^{S457D}$. (D) Growth tests of low-density cell suspensions on SLAD and SHAD media at day 3 (3d) and 7 (7d). The homozygous diploid triple-*mepΔ* growth tests showed in panel A served as common control for tests A and D. (E) Pseudohyphal growth tests of high-density cell suspensions dropped on SLAD and SHAD media at day 7.

between TM helices 5, 10 and 11. In ScMep2, residue H199 is located in the vicinity of the conserved H194-H348 histidine-twin (His1-His2) lying along the conducting pore and playing a key role in transport in Mep-Amt-Rh proteins (Fig 5A and 5B) [17, 18, 28–30, 35, 42–44]. The H199Y substitution thus circumvents the requirement of the CTD to enhance substrate translocation through the hydrophobic core of Mep2. Mep2$^{H199Y, S426stop}$ and its equivalent version comprising the CTD, Mep2$^{H199Y}$, both enabled efficient complementation of the growth defect of triple-*mepΔ* cells on both SLAD and SHAD (Fig 4B). Importantly, despite the absence of the whole CTD, Mep2$^{H199Y, S426stop}$ turned able to allow filamentation on SLAD, although with a reduced efficiency compared to native Mep2 (Fig 4A). Cells expressing Mep2$^{H199Y, S426stop}$ showed a slightly increased capacity to remove ammonium compared to native Mep2 when growing exponentially in SHPD (Fig 4C). In the latter case, the correlation between transport and filamentation efficiency appeared not supported. However, these data reveal that the CTD of Mep2 is not strictly required to allow filamentation, while they do not exclude that it somehow plays a positive role in this process. Indeed, the Mep2$^{H199Y}$ variant conserving the CTD induced hyperfilamentation on SLAD compared to Mep2, and some filaments even appeared on SHAD (Fig 4A). The variant also showed a slightly increased efficiency of ammonium removal in SHPD (Fig 4C), in this case in line with a link between transport and signaling capacities.

As the cellular context of Npr1-lacking cells appears to influence the Mep2 activity in CTD-dependent and independent ways, we checked the behavior of the H199Y variant in triple-*mepΔ npr1-1* cells. The Mep2$^{H199Y}$ variant was able to suppress the growth defect on SLAD and SHAD (Fig 4B), indicating that the H199Y substitution circumvents the requirement of Npr1 for Mep2 activity. Remarkably, the Mep2$^{H199Y, S426Stop}$ variant allowed more filamentation in triple-*mepΔ npr1-1* cells than in the presence of Npr1, further indicating that the CTD is dispensable for induction (Fig 4A). The Mep2$^{H199Y}$ variant conserving the CTD induced hyperfilamentation on SLAD also in the absence of Npr1 (Fig 4A).

We characterized another mutation in the pore that was isolated for its ability to confer activity to Mep2 in Npr1-lacking cells [25]. The G349C mutation lies next to H348 of the histidine-twin in the 10$^{th}$ TM α-helix. We show that, as H199Y, the G349C substitution conferred activity to a Mep2 protein lacking the CTD. When growing exponentially in SHPD, triple-*mepΔ* cells expressing Mep2$^{G349C, S426stop}$ showed a slightly increased capacity to remove ammonium (0.5 mM) from the extracellular medium compared to native Mep2, while the Mep2$^{G349C}$ variant conserving the CTD was even more efficient (Fig 4D). A growth advantage with the Mep2$^{G349C}$ variant was visible on SHAD (1 mM ammonium) at day 7 (Fig 4B), while this advantage was not observed on SLAD (0.1 mM ammonium), probably due to an altered affinity for ammonium. Although Mep2$^{G349C, S426stop}$ was able to support growth of triple-*mepΔ* cells and triple-*mepΔ npr1-1* cells on SLAD and SHAD (Fig 4B), only the variant with the CTD turned able to allow filamentation on SLAD in triple-*mepΔ* cells (Fig 4A). In contrast, in Npr1-lacking cells, Mep2$^{G349C, S426stop}$ also allowed filamentation on SLAD, supporting that the CTD can be dispensable for induction (Fig 4A). Like the Mep2$^{H199Y}$ variant, Mep2$^{G349C}$

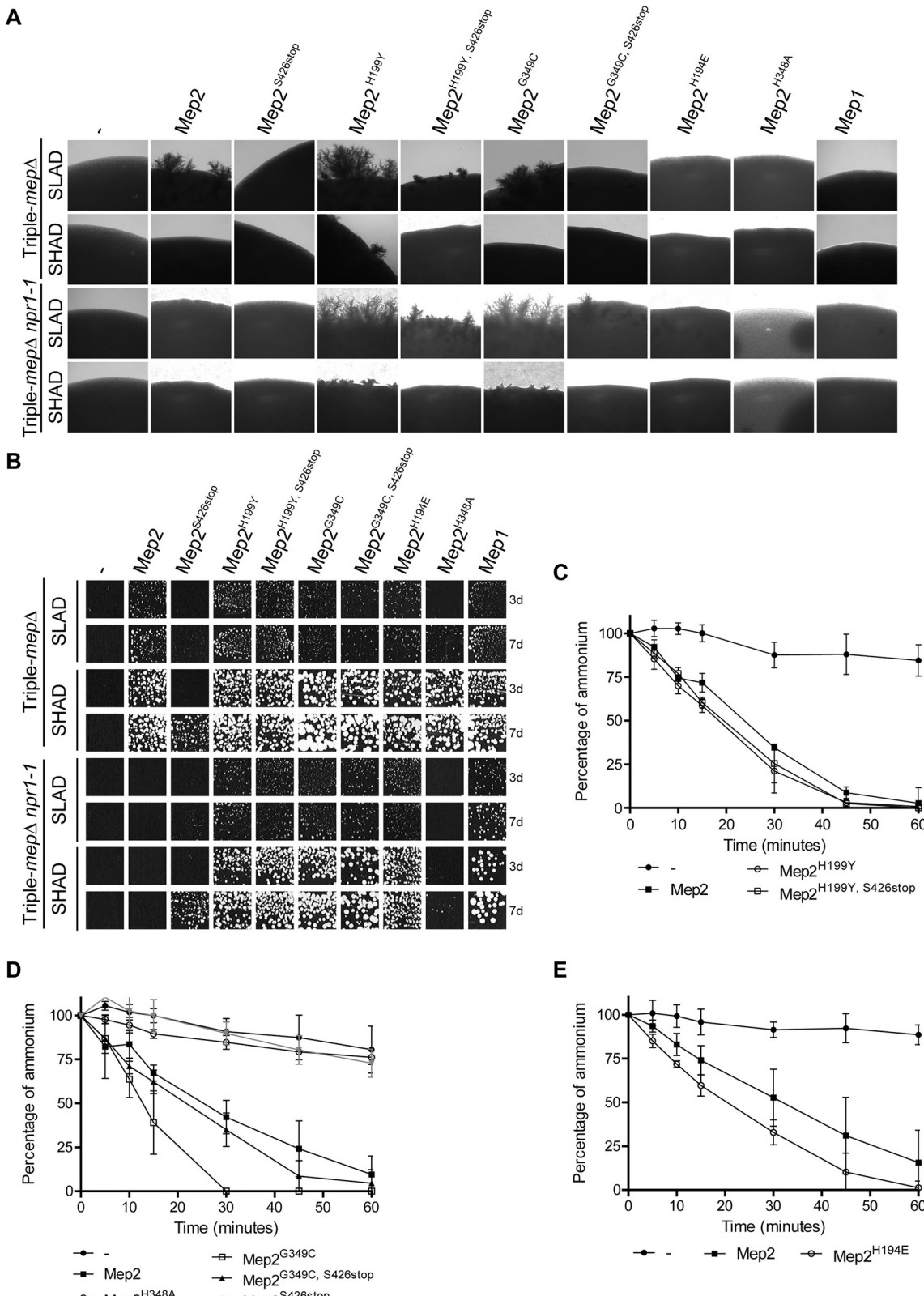

**Fig 4. The Mep2 C-terminal tail can be dispensable for filamentation induction.** (A-B) Homozygous diploid triple-*mepΔ* (ZAM38) and triple-*mepΔ npr1-1* (ZMB058) cells were transformed with the pFL38 empty plasmid (-) or with YCpMep2, YCpMep2^S426stop,

YCpMep2[H199Y], YCpMep2[H199Y, S426stop], YCpMep2[G349C], YCpMep2[G349C, S426stop], YCpMep2[H194E], YCpMep2[H348A] or YEpMep1. (A) Pseudohyphal growth tests of high-density cell suspensions dropped on SLAD and SHAD media at day 7. (B) Growth tests of low-density cell suspensions on SLAD and SHAD media at day 3 (3d) and 7 (7d), at 29˚C. (C-E) Ammonium removal rates of homozygous diploid triple-*mepΔ* (ZAM38) cells growing in SHPD (0.1% proline) liquid medium. At time 0, 500 μM ammonium was added and its removal from the medium was monitored for 1 h. Ammonium remaining in the medium is expressed as percentage of the initial concentration. Averages and standard deviations are reported (n = 3). (C) Cells transformed with the pFL38 empty plasmid or with YCpMep2, YCpMep2[H199Y], YCpMep2[H199Y, S426stop]. (D) Cells transformed with the pFL38 empty plasmid or with YCpMep2, YCpMep2[H348A], YCpMep2[G349C], YCpMep2[G349C, S426stop], YCpMep2[S426stop]. (E) Cells transformed with the pFL38 empty plasmid or with YCpMep2, or YCpMep2[H194E].

conserving the CTD induced hyperfilamentation on SLAD (Fig 4A) [25]. Filamentation tips were also visible on SHAD (Fig 4A).

## H199 and G349 are face-to-face near the pore histidine-twin and TM11 of Mep2

We examined the 3D environment of residues H199 and G349 in the X-ray structure of ScMep2 (Fig 5A) [15]. Both residues, respectively located in TM5 and TM10, are face-to-face on the helix-helix packing. The side chain of H199 is sandwiched between Y345 on one side and, on the other side, the pair of aliphatic residues L401 and V405 both located on TM11, the last long helix that surrounds the lipid-accessible side of each monomer and that precedes the CTD (Fig 5C and 5D). A T-shaped π-π interaction is observed between aromatic rings of H199 and Y345. The H199 side chain makes also van der Waals contacts with both side chains located nearly one helical turn on both sides, i.e. L195 in α-helix upstream and L202 downstream. Based on the crystal structure, H199Y and G349C mutations were predicted as having, respectively, little and destabilizing effects on the thermal protein stability (S2 Table). Experimental data showed that both Mep2[H199Y] and Mep2[G349C] conserve transport activity and the last is as abundant as native Mep2 (Fig 4B–4D) [25], indicating that the proteins are stable and that structural rearrangements probably occur to accommodate these substitutions.

We further compared the sequences of hundreds of fungal Mep1-like and Mep2-like proteins in order to highlight potential specific peculiarities and to establish the degree of conservation of a given residue (S2 Fig). G349 in ScMep2 is very well conserved amongst both Mep1 and Mep2-like proteins, alanine being the other alternative residue encountered at this position (Fig 5B). In contrast, the nature of the residue at position H199 is variable amongst Mep2-like and Mep1-like proteins. This position is most frequently occupied by A, T, S and H in Mep2-like proteins whereas V, M and L are often found in Mep1-like proteins (Fig 5B). Only one analyzed Mep2-like sequence had a Y on the related position (0.29%), H being found in 12.64% of the cases, and F in 8.62%, including CaMep2.

Many amino acids are fully conserved across Mep1 and Mep2-like proteins and spread over the entire protein sequence (S2 Fig). The most obvious difference between Mep2-like and Mep1-like proteins is clearly the absolute preservation of an histidine residue at position His1 of the histidine-twin in Mep2-like, H194 in ScMep2, whilst this position is invariably occupied by a glutamate in Mep1-like proteins, E181 in ScMep1 [29]. Other differences, though less obvious, may nevertheless be observed, such as F243(ScMep2) vs L226(ScMep1), I303 (ScMep2) vs A285(ScMep1), T359(ScScMep2) vs N341(ScMep1) and to a lesser extent T172 (ScMep2) vs V159(ScMep1), T229(ScMep2) vs S212(ScMep1) and N356(ScMep2) vs L338 (ScMep1). In addition, F281(ScMep2) has no counterpart in ScMep1. While the Mep2 cytoplasmic enhancer domain (region 428–441 in ScMep2) displays strong sequence similarity with Mep1-like proteins, the autoinhibitory domain of Mep2 is largely distant from Mep1 proteins (S2 Fig).

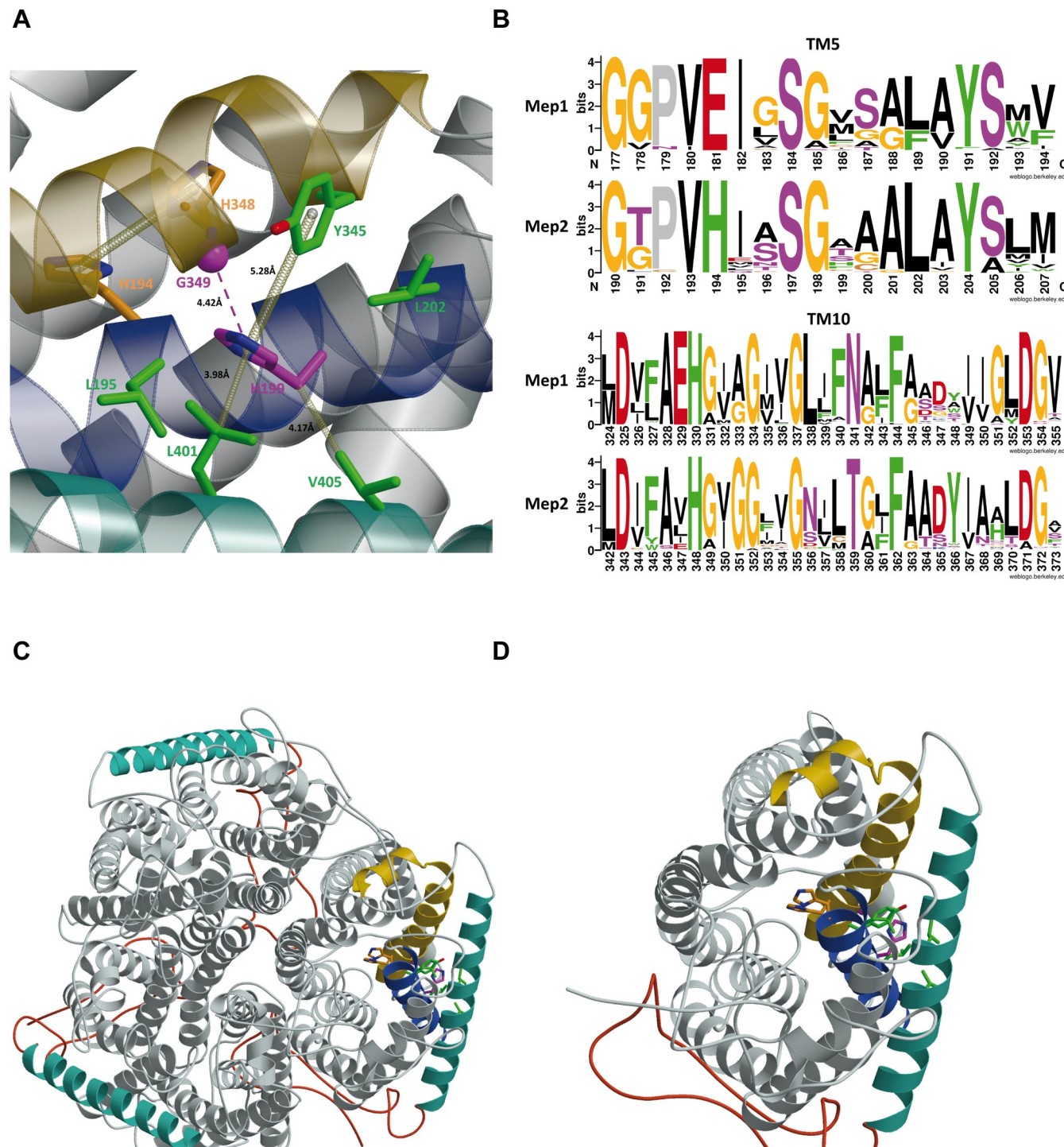

**Fig 5. Visualization of interatomic interactions implicating H199 and G349 residues in the ScMep2 crystal structure.** A close-up (A), a global trimeric (C) and a monomer (D) view of the ScMep2 structure (PDB id 5AEX, chain A). ScMep2 protein is illustrated in grey ribbons with TM5 in blue, TM10 in yellow and TM11 in turquoise, and with the C-terminal coil in red. The H199 side chain is depicted in stick representation with carbon atoms in purple and the α carbon atom of G349 is drawn as a purple sphere. Side chains found in interaction with H199 are shown in stick representation with carbon atoms in green. The histidine-twin H194-H348 is also depicted with carbon atoms in orange. The nitrogen and oxygen atoms are colored in blue and red, respectively. In (A), different type of non-covalent interactions are illustrated by yellow dashed bonds as π-π stack interactions and van der Waals interactions. The distances associated with non-covalent interactions are indicated, as well as the distance between Cα of G349 and ring of H199. Residues drawn are labelled.

The non-covalent interactions were defined with the ARPEGGIO program and the images were created using a combination of MolScript and Raster3D programs. (B) WebLogo plots of transmembrane helices TM5 and TM10 in Mep1-like vs Mep2-like sequences. The logo consists of amino acid stack for each position in the aligned sequence dataset. The height of the stack indicates the sequence conservation at that position (maximum = 4 bits), while the height of the amino acid one-letter symbol within the stack shows the relative frequency of the amino acid at that position. The color scheme is aromatic amino acids (F, H, W, Y) in green, polar (N, Q, S, T) in purple, negatively charged (D, E) in red, positively charged (K, R) in blue, aliphatic (A, I, L, M, V) in black, cysteine (C) in brown, proline (P) in grey and glycine (G) in orange. Sequence numbering is according ScMep1 or ScMep2 sequences.

## The Mep2 pore histidines are not equally impacting on filamentation induction

We previously described that the H194E mutation at position His1 of the histidine-twin preserves activity of yeast Mep2 while it switches the kinetic properties to ones close to those of Mep1 and leads to loss of filamentation capacity [29]. We consequently proposed the existence of two functional groups of fungal Mep proteins showing differences in the transport mechanism and filamentation induction abilities. Here, we further show that Mep2$^{H194E}$ allowed ammonium removal by triple-*mepΔ* cells in SHPD even faster than native Mep2 (Fig 4E). In addition, while Mep2$^{H194E}$ was also able to suppress the growth defect of triple-*mepΔ npr1-1* cells on SLAD and SHAD (Fig 4B), it was still unable to allow filamentation (Fig 4A). The absence of filamentation was also observed when the variant was expressed in a simple *mep2Δ* background (S1 Fig). Hence, transport and signaling functions are uncoupled in Mep2$^{H194E}$. A specific substrate translocation mechanism via the Mep2 pore is connected to filamentation capacity.

The position His2 of the histidine-twin is invariably occupied by a histidine in all Mep-Amt-Rh ammonium transport proteins, including in Mep1-like and Mep2-like proteins. The H348A substitution of the His2 has also been reported to preserve activity while impairing the filamentation capacity of Mep2 [30]. Indeed, Mep2$^{H348A}$ did not allow filamentation of *triple-mepΔ* cells (Fig 4A). However, the variant did not efficiently complement the growth defect of the latter cells streaked on SLAD (Fig 4B), nor was it able to allow ammonium removal in SHPD (Fig 4D). These data indicate that the variant is affected in its transport activity therefore precluding any conclusion on its capacity to induce filamentation. Absence of filamentation was observed when Mep2$^{H348A}$ was expressed in a simple *mep2Δ* background, condition where growth is ensured by ammonium uptake via Mep1 and Mep3 (S1 Fig). Mep2$^{H348A}$ did not suppress the growth defect of triple-*mepΔ npr1-1* cells, even if a few suppressor colonies appeared on SHAD after prolonged growth (Fig 4B). Therefore, in the case of the H348A variant, absence of filamentation induction is correlated to a reduction of transport activity rather than to an uncoupling of transport and signaling functions. These data further support a correlation between transport and filamentation efficiencies.

As for several Mep2 variants, we observed an increased filamentation efficiency in cells lacking Npr1, we evaluated whether the kinase absence could *per se* enhance filamentation, independently of Mep2 in triple-*mepΔ npr1-1* cells. We previously showed that overexpression of Mep1 in Npr1-lacking cells is sufficient to bypass the kinase requirement and allow Mep1 activity [40]. Consistently, Mep1 overexpression enabled growth of triple-*mepΔ npr1-1* cells on SLAD and SHAD (Fig 4B). However, no filamentation was observed, indicating that substrate translocation via the pore of Mep2 but not of Mep1 allows filamentation in Npr1-lacking cells (Fig 4A).

Overall, the data indicate that the CTD of Mep2 can be dispensable for filamentation induction and that substrate must be translocated via the Mep2 pore following a specific mechanism to allow filamentation induction.

## Substrate translocation via Mep1 and Mep2 has different impact on cytosolic pH

A number of studies support that Mep-Amt-Rh proteins translocate $NH_3$ after initial recognition and deprotonation of the $NH_4^+$ entity [18, 34, 36, 37, 45, 46]. Therefore we tried to determine the *in vivo* impact of substrate transport via either Mep1 or Mep2 on the cytosolic pH ($pH_c$) using a defined growth condition previously shown to enable the recording of both Mep activities within minutes [11, 14]. Indeed, using buffered minimal medium (pH 6.1) with proline as nitrogen source, Mep1 and Mep2 independently ensure rapid $^{14}C$-methylammonium accumulation. Here, to measure cytosolic $pH_c$ variation during the substrate translocation process, we used the cytosolic reporter pHluorin, a ratiometric variant of GFP [47]. As expected, addition of ammonium (2 mM) had no impact on the $pH_c$ of triple-*mepΔ* cells consistent with the inability of these cells to transport ammonium at low concentrations (Fig 6A). While triple-*mepΔ* cells are unable to grow in the presence of ammonium concentrations below 5 mM (pH 6.1), higher ammonium concentrations enable growth, likely due to a sufficient passive diffusion of the neutral form $NH_3$ [11]. Accordingly, addition of ammonium at 20 mM to triple-*mepΔ* cells induced a slight increase of $pH_c$ (Fig 6A). In wild-type cells, ammonium addition (2 mM) induced a multiphasic response, with a transient $pH_c$ drop and a $pH_c$ recovery phase followed by an alkalinisation phase (Fig 6B). pH maintenance in fungi is ensured by two main key players, the vacuolar $H^+$ V-ATPase and the plasma-membrane $H^+$-ATPase Pma1, coping with cytosolic acidification [48]. Extrusion of protons via Pma1 is likely responsible for the $pH_c$ recovery phase in response to the initial pH drop. Cells expressing only Mep1 showed an initial transient drop of $pH_c$, but after the $pH_c$ recovery phase, the alkalinisation phase compared to initial $pH_c$ was less marked than in wild-type cells (Fig 6C). Cells expressing only Mep2 had a different behavior, the initial acidification phase was less marked while the alkalinisation phase was observed as in wild-type cells (Fig 6D). The wild-type behavior appears to recapitulate a combination of the Mep1 and Mep2 behaviors (Fig 6E).

As the H194E mutation appears to alter the mechanism of ammonium translocation via Mep2, we determined the pH change associated with substrate transport via the Mep2$^{H194E}$ variant. For that purpose, Mep2 variants directly fused to pHluorin in their C-terminal extremity were expressed in triple-*mepΔ* cells. This technique shows the advantage of reporting local pH variations near the plasma membrane, close to Mep2 proteins. Fluorescence microscopy confirmed the plasma membrane localization of the tested Mep2-pHluorin variants (Fig 7). In proline-grown cells expressing the inactive Mep2$^{D186N}$-pHluorin, addition of ammonium (2 or 0.5 mM) had no clear impact on the pH, consistent with the inability of Mep2$^{D186N}$ to recognize and transport ammonium (Fig 7B). In cells expressing Mep2-pHluorin, ammonium addition, either at 2 or 0.5 mM, induced an instant and transient pH drop followed by a rapid alkalinisation phase (Fig 7A). In cells expressing Mep2$^{H194E}$-pHluorin, addition of 2 or 0.5 mM ammonium induced a pronounced pH drop followed by a pH recovery phase but no alkalinisation phase compared to initial pH (Fig 7C). The drop was even more important than the one observed for $pH_c$ in cells expressing Mep1 (Fig 6C).

These data indicate that, in the tested conditions, Mep1 and Mep2-mediated translocations of substrate(s) have specific different impacts on $pH_c$. Transport activity of Mep2$^{H194E}$ appears to impact intracellular pH in a similar way than Mep1 activity, still suggesting that the H194E mutation modifies the transport mechanism of Mep2 into a transport mechanism closer to the one of Mep1.

## Substrate translocation via Mep1 and Mep2 operates via distinct mechanisms

Electrophysiology analyses were next performed upon heterologous expression of Mep1 and Mep2 proteins in *Xenopus* oocytes. Using ammonium 3 mM, currents were recorded for

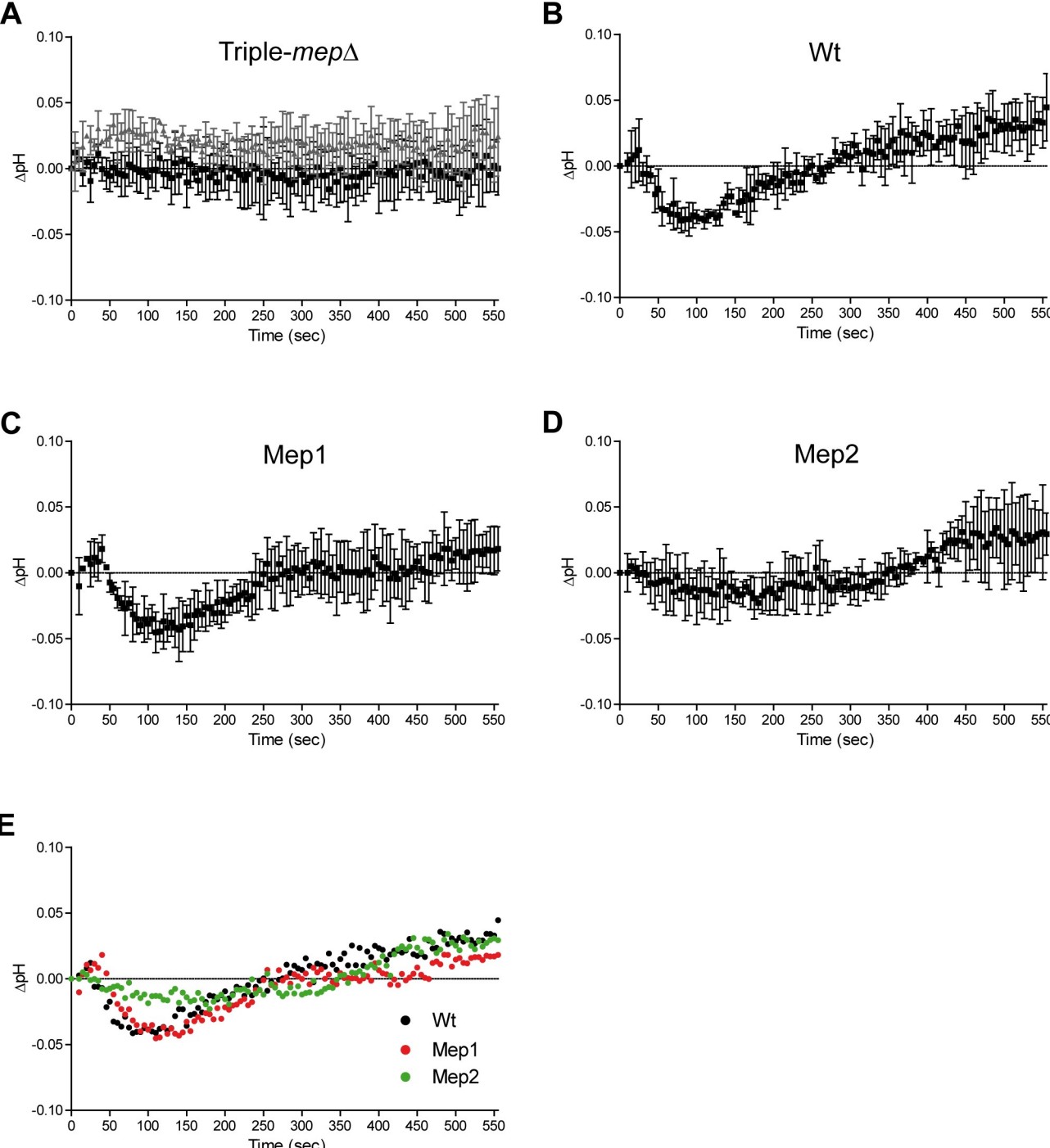

**Fig 6. Substrate translocation via Mep1 and Mep2 has specific impacts on cytosolic pH.** Cells transformed with pYES-pHluorin were grown in buffered minimal medium (pH = 6.1) containing 0.1% proline and evolution of pH$_c$, corrected for negative control variations (delta-pH), was followed after addition of ammonium 20 mM (grey triangle) (A) and/or 2 mM (black square) (A-D). (A) Triple-*mepΔ* (31019b) cells. (B) Wild-type (23344c) cells. (C) *mep2Δ mep3Δ* (31018b) cells, expressing chromosomal *MEP1*. (D) *mep1Δ mep3Δ* (31022a) cells, expressing chromosomal *MEP2*. (A-D) Averages and standard deviations are reported (n = 3). (E) Combined representation of delta-pH evolution from (B-D) after addition of 2 mM ammonium. Wild-type (black circle), *mep2Δ mep3Δ* (red circle) and *mep1Δ mep3Δ* (green circle) cells. Averages are reported (n = 3).

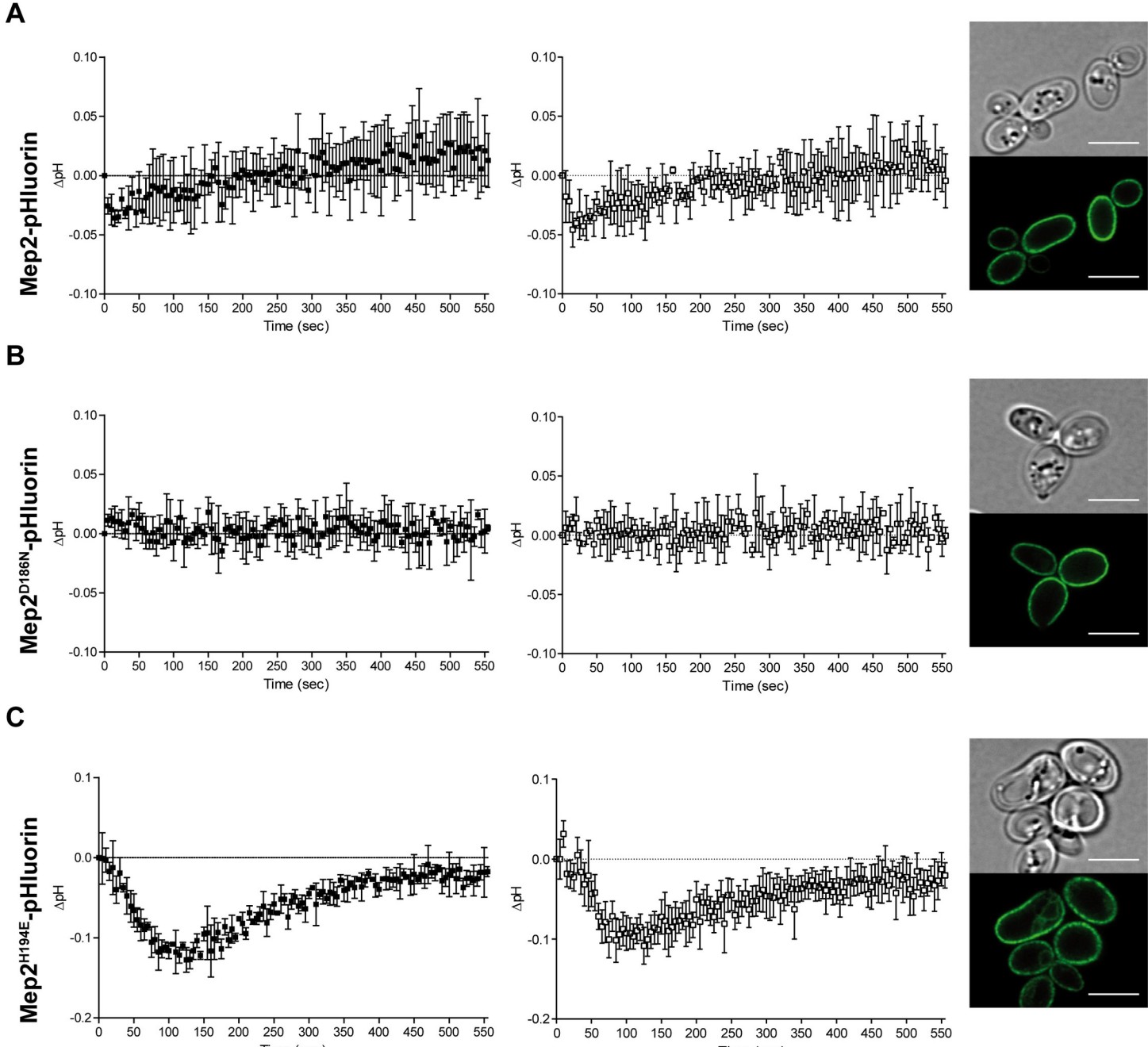

**Fig 7. Substrate translocation via Mep2$^{H194E}$ induces a pH decrease.** Haploid triple-*mepΔ* cells (31019b) were transformed with pMep2-pHluorin (A), pMep2$^{D186N}$-pHluorin (B) and pMep2$^{H194E}$-pHluorin (C) and grown in a buffered minimal medium (pH = 6.1) containing 0.1% proline. Evolution of pH, corrected for negative control variations (delta-pH), was followed after addition of ammonium 2 mM (black square) and 0.5 mM (open square). The cells were also visualized by fluorescence microscopy (panel 3). Scale bar, 5 μm.

Mep1, saturating with a voltage-dependent $K_m$ of 258 μM at -100 mV (Fig 8A–8C). By contrast, no current was recorded for Mep2 (Fig 8A and 8B). The phosphomimetic S457D substitution did not favor Mep2 activity in the tested conditions. In addition, the H194E substitution with or without the combination of S457D had no impact on the recorded

currents (Fig 8A and 8B). We tested whether the absence of current was due to mislocalization, potentially linked to a Mep2 glycosylation matter. GFP-tagged versions of Mep2 and of the non-glycosylable Mep2^N4Q variant were both targeted to the oocyte surface (Fig 8D), indicating that glycosylation is not representing an issue for correct localization in oocyte, similarly to the yeast system [19]. Absence of current with Mep2^S457D was also not due to mislocalization of the protein. By contrast, the H194E substitution affected the stability of the GFP-tagged Mep2 protein, indicating that we cannot conclude about the transport mechanism used by this variant expressed in oocytes (Fig 8D). Finally, $^{15}$N-ammonium uptake experiments were performed to evaluate the functionality of the Mep proteins in oocytes. Of note, our data reveal that Mep2 and its Mep2^S457D variant were functional in oocytes, enabling substrate transport as well as Mep1 (Fig 8E).

Hence, in the tested conditions, substrate translocation is electrogenic via Mep1 whilst it is electroneutral via Mep2, supporting a different transport mechanism between these proteins.

## The H188E pore mutation abolishes signaling ability of *C. albicans* Mep2 in *S. cerevisiae*

The orthologue of *S. cerevisiae* Mep2 from *C. albicans*, CaMep2, is likewise required for filamentation of the pathogen whilst its CaMep1 paralogue is not [22]. Only CaMep2 is able to allow filamentation upon heterologous expression in *S. cerevisiae* [22]. This suggests that ScMep2 and CaMep2 employ a similar signaling mechanism. Like in *S. cerevisiae*, CaMep2 displays the conserved histidine-twin in the conducting pore while in CaMep1 a glutamate replaces the first histidine residue of the pair. We therefore addressed the impact of the H188E substitution, the equivalent of H194E in ScMep2, on the signaling function of CaMep2 in *S. cerevisiae*. Vectors were constructed to express and overexpress CaMep2 and CaMep2^H188E, under the control of the Sc*MEP2* promoter and terminator, in triple-*mepΔ* cells. CaMep2 and CaMep2^H188E expressed from a centromeric plasmid did not efficiently complement the growth defect of triple-*mepΔ* cells, growth being only barely visible after 7 days on SHAD (Fig 9A). No filamentation was observed in these conditions (Fig 9B). The same variants expressed from an episomal plasmid allowed a better complementation of triple-*mepΔ* cells, growth being visible after 7 days on both SLAD and SHAD (Fig 9A). Filamentation was observed on SLAD and SHAD with cells overexpressing CaMep2 but not CaMep2^H188E (Fig 9B). Of note, overexpressed CaMep2^H188E allowed efficient growth on SHAD but not filamentous growth indicating that the H188E mutation impairs signaling.

Overall, the histidine-to-glutamate pore substitution abolishes the signaling ability of both ScMep2 and CaMep2 indicating that a conserved transport mechanism might underlie their filamentation signaling capacity in baker yeast and also in *C. albicans*.

## Discussion

This work brings novel insights into the molecular mechanism enabling yeast Mep2-type ammonium transport proteins to trigger the signal leading to filamentation induction. Overall, our data support that the signaling property of Mep2 is intimately linked to the molecular events occurring during substrate translocation through the pore. Our interest was first focused on the role of the Mep2 CTD in filamentation induction as we previously showed this region plays an important role in fine-tuning the transport activity of the hydrophobic core of the protein [39]. The C-terminal extension is also the largest part of the protein facing the cytosol, the N-terminus being periplasmic [19]. Mep2 cytosolic domains are likely to interact with signaling partners in analogy to 7TM GPCRs that are widely appreciated to act as platforms coordinating interactions of proteins involved in a variety of aspects of cell signaling [49].

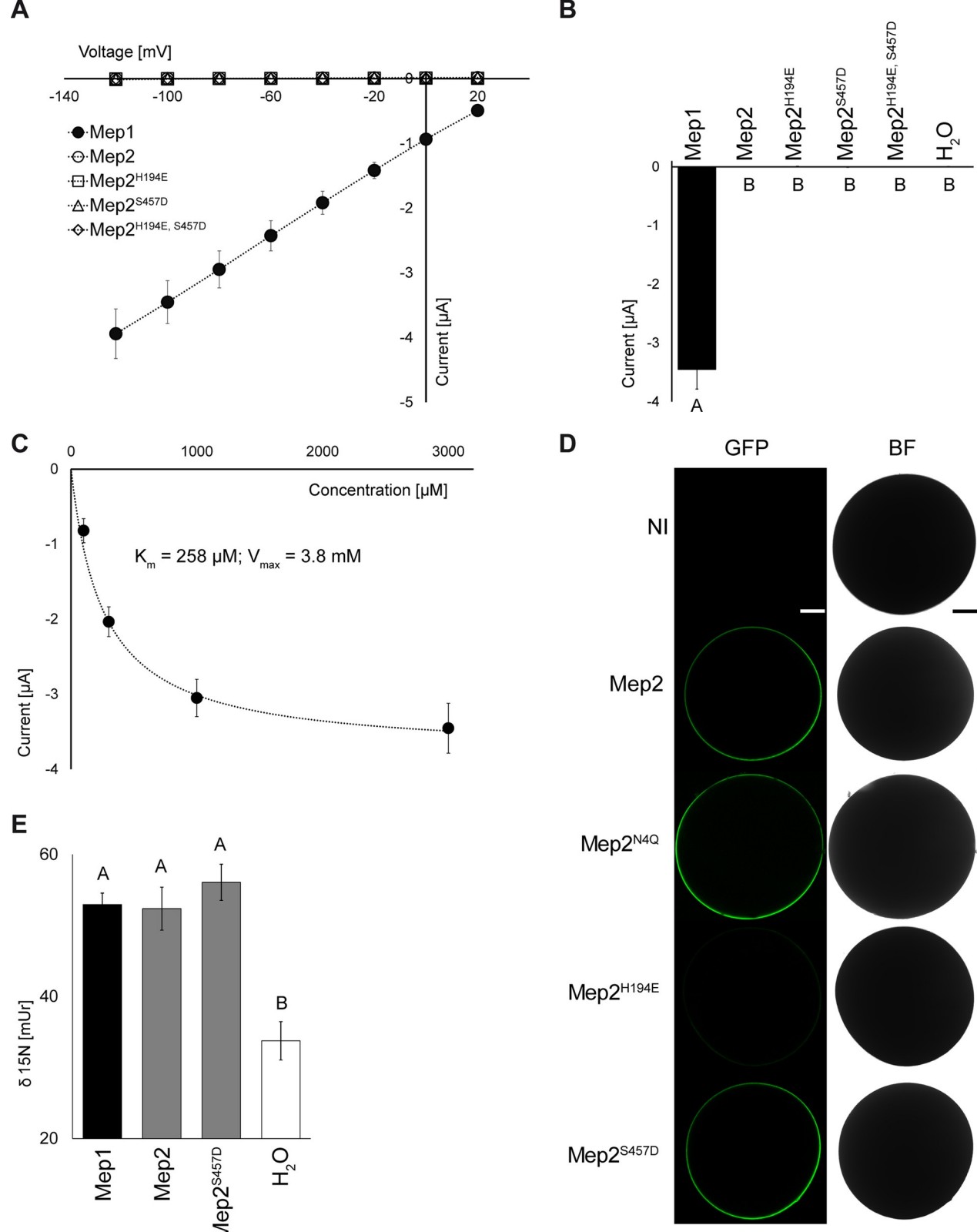

**Fig 8. Functional characterization of Mep1 and Mep2 in *Xenopus* oocytes.** (A) Current / Voltage plot showing currents induced by 3 mM ammonium in dependence of the membrane potential. Mep1 is shown as closed black circles. Mep2, Mep2$^{H194E}$, Mep2$^{S457D}$ and Mep2$^{H194E\ S457D}$ are

shown as open, circles, squares, triangles and diamonds, respectively. All currents were recorded at pH 5.5 from the same batch of oocytes. Data is given as means (n≥4) ± SD. (B) Currents induced by 3 mM NH$_4$Cl at a membrane potential of -100 mV. Data is shown as means ± SE (n≥4) and significant differences at P<0.01 according to Tukey's test are indicated by different letters. (C) Concentration-dependence of the ammonium currents of Mep1 at pH 5.5. Net currents at -100 mV are given in dependence of the ammonium concentration. Data is given as means ± SD (n = 5). (D) Mep2-GFP, Mep2$^{N4Q}$-GFP, Mep2$^{H194E}$-GFP and Mep2$^{S457D}$-GFP fusion protein localization at the plasma membrane of oocytes. Control shows non-injected (NI) oocytes. Dark pictures show GFP fluorescence in the membrane, bright pictures show bright-field (BF) images of the oocyte. Shown is one representative picture of n = 30. Scale bar, 200 μm. (E) Influx of $^{15}$N-labelled ammonium into oocytes expressing Mep1, Mep2, and Mep2$^{S457D}$. Water injected oocytes were used as negative control. Oocytes were incubated for 20 min in media containing 3 mM NH$_4$Cl at pH 5.5. Data is given as means ± SE (n≥8) and significant differences at P<0.01 according to Tukey's test are indicated by different letters.

A close link was observed between the efficiency of filamentation induction and the substrate transport efficiency of Mep2 that impacts on the consequent growth ability at limiting ammonium levels. Mep2 proteins lacking the autoinhibitory CTD enabled improved substrate transport and supported enhanced filamentation while those bearing truncations in the enhancer domain showed reduced transport capacity and did not support efficient growth of triple-*mepΔ* cells nor filamentation induction. The transport and signaling functions of Mep2 are thus not fully uncoupled in the latter variants. This conclusion contrasts with the behavior of a CTD variant of the Mep2-type protein from *Candida albicans* affected in the corresponding enhancer region. The respective variant did not allow filamentation while ensuring growth of a double-*mep* mutant strain and enabling ammonium removal at levels comparable to native Mep2 [22]. Variants of *C. albicans* and *S. cerevisiae* Mep2 orthologues frequently behave differently. For instance, the CaMep2$^{D180N}$ variant equivalent to ScMep2$^{D186N}$ affected in substrate recognition is unstable and mislocalized [28, 31]. Further, the G349C mutation in ScMep2 suppresses the Npr1 kinase requirement for activity and allows filamentation induction, while the corresponding counterpart CaMep2$^{G343C}$ turns similarly insensitive to CaNpr1 but loses the ability to allow filamentation, possibly caused by a reduced ammonium transport efficiency [25, 50]. Specificities in experimental procedures (medium, temperature, agar) might potentially delineate some discrepancies. In our studies, *S. cerevisiae* cells are grown at 29˚C in all experiments, while in *C. albicans* the dependency of CaMep2 activity and

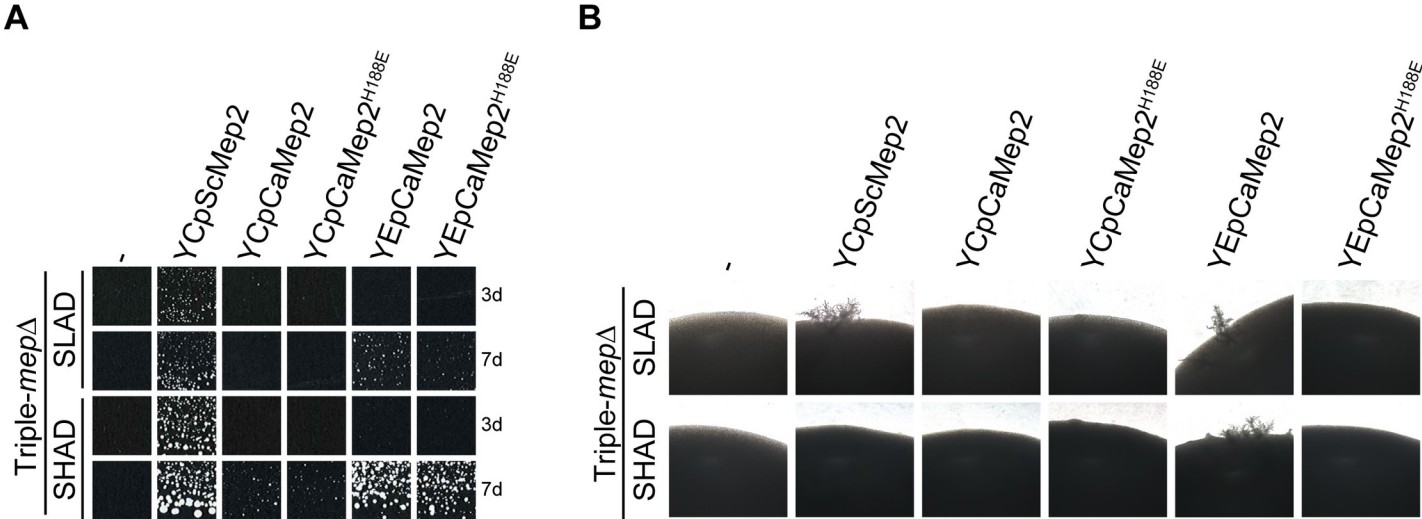

**Fig 9. The H188E pore mutation abolishes the signaling ability of *C. albicans* Mep2 expressed in *S. cerevisiae*.** Homozygous diploid triple-*mepΔ* (ZAM38) cells were transformed with the pFL38 empty plasmid (-) or with the centromeric plasmids, YCpMep2, YCpCaMep2, YCpCaMep2$^{H188E}$, or with the episomal plasmids, YEpCaMep2 or YEpCaMep2$^{H188E}$. (A) Growth tests of low-density cell suspensions on SLAD and SHAD media at day 3 (3d) and 7 (7d) at 29˚C. (B) Pseudohyphal growth tests of high-density cell suspensions dropped on SLAD and SHAD media at day 7 at 29˚C.

filamentation induction on the CaNpr1 kinase observable at 30˚C is lost at 37˚C, indicating that a raise in the temperature is allowing a conformation permissive for transport despite the absence of the kinase [50]. In *S. cerevisiae*, single substitutions in periplasmic, hydrophobic or cytosolic domains including intracellular loops and CTD are able to confer a permissive conformation to Mep2 allowing activity even in the Npr1 absence [39]. Thus, minor changes all along Mep2 can suppress the Npr1 requirement for activity and negative control by the autoinhibitory domain. Here, on the filamentation-induction medium, a number of CTD Mep2 variants including the one lacking the autoinhibitory domain allowed increased growth and filamentation when expressed in Npr1-lacking cells indicating that the cellular context in the absence of the kinase also influences the Mep2 activity in a manner independent of the autoinhibitory CTD. On one hand, the apparent increased abundance detected by western blot for slightly active Mep2 variants could reflect an *in vivo* increased abundance that could contribute to the growth and filamentation observed in the absence of Npr1. Derepression of genes submitted to the nitrogen catabolite repression has indeed been reported in *npr1* cells [51]. However, a better recognition efficiency of less phosphorylated Mep2 proteins by the anti-peptide polyclonal antibodies used in western blots could also contribute to the apparent increased abundance observed in *npr1* cells. In addition, we found that the $pH_c$ of Npr1-lacking cells (strain 30788a) is significantly more acidic than the one of 23344c wild-type cells (6.96±0.08 vs 7.10±0.08, $p<10^{-5}$, n = 19); this might lead to rearrangements in Mep2 allowing increased activity or favoring the substrate translocation. In the same line, we here observed a better complementation efficiency of CTD Mep2 variants in the YNB-based unbuffered medium, where the pH can easily drop below pH 5 during growth [51], compared to the minimal buffered medium (pH 6.1) used in a previous study [39]. This likely relies on the optimal pH of transport which is close to pH 4.5 for Mep2 [29].

In the scope to design a Mep2 variant showing uncoupled transport and signaling properties, we constructed a transport-inactive Mep2$^{D186N, S457D}$ variant with mutations favoring an active structural conformation of Mep2 [15, 39], while preventing substrate recognition and therefore translocation [31]. Together our data indicate that substrate must be recognized, or that Mep2 must undergo the conformational changes occurring during substrate translocation to allow filamentation. Signaling usually starts by the binding of a ligand to a weakly expressed receptor triggering a signal amplified by second messengers. A peculiarity of the Mep2-dependent signaling resides in the requirement of high expression levels of the transport protein to trigger filamentation [22], suggesting that signal amplification occurs at the very first step of the transmission. Under non-preferred nitrogen source supplementation, Mep2 is an abundant protein, far more abundant than Mep1 or Mep3, while its $V_{max}$ is the lowest amongst the three Meps suggesting that the turnover of Mep2 is much lower than Mep1 and Mep3 [11]. Moreover, engineered overexpression of Mep2-type proteins is sufficient to trigger inappropriate filamentation during sufficient nitrogen supply [30, 52]. These observations suggest that there is a quantitative requirement for sufficient substrate translocation events via Mep2 to trigger filamentation. This is also supported by the filamentation occurring even on SHAD only when *C. albicans* Mep2 is overexpressed in *S. cerevisiae* cells. In baker yeast, the Mep2$^{CΔ428–449}$ and Mep2$^{CΔ434–485}$ variants are able to induce efficient filamentation in Npr1-lacking cells despite internal deletions in sum covering large parts of the CTD. In these variants, the region corresponding to the one shown as important for filamentation but dispensable for transport in *C. albicans*, is either totally or largely removed [22]. The behavior of Mep2$^{H199Y, S426stop}$ and Mep2$^{G349C, S426stop}$ indicates that, if the translocation event is *per se* able to trigger the signal, the latter can be transmitted despite the absence of the CTD. However, the CTD seems to improve the signaling process, as these truncated proteins are not as efficient as native Mep2 in enabling filamentation. The CTD might for instance contribute to

the stability of the protein. Substrate translocation process through the pore of the hydrophobic core appears nevertheless sufficient to allow filamentation. If potential signaling partners exist, they do not exclusively bind to the CTD. Interaction could also take place at the level of the intracellular loops or laterally in the lipid bilayer.

At the transport function level, H199Y and G349C substitutions suppress the requirement of the C-terminal tail to upregulate substrate translocation through the pore of Mep2. The relative position of both residues, their vicinity to the key histidine-twin of the pore and the network formed with I401 and V405 residues in the TM11 suggest that these residues participate to the mechanism enabling the C-terminal tail to fine-tune transport. TM11 crosses diagonally the lipid-exposed face of one monomer to end in the cytoplasm where it is directly prolonged by the CTD. The substitutions may have induced a repositioning of the TM11 domains surrounding the trimer and promoted a conformation influencing the position of the pore histidines that facilitates transport. In *E. coli* AmtB, a substitution to leucine of I359 which lies at one helix turn from the corresponding I401 in ScMep2 enables to bypass the C-terminus requirement [53]. As the side chain of I359 contacts residues in TM5 and TM10, this could be consistent with TM11 influencing the positions of the critical histidine-twin, and therefore transport.

One major difference between the pores of Mep2-type and Mep1/3-type proteins is the invariable presence in the latter of a glutamate at the position His1 of the otherwise conserved histidine-twin found in Mep-Amt-Rh ammonium transport proteins. We have shown that a Mep1 variant with a reconstituted histidine-twin is inactive, indicating that histidine and glutamate are not equivalent in the pore [29]. Inversely, a Mep2$^{H194E}$ variant with a glutamate at His1 is active but shows important modifications in the kinetic of substrate translocation, with an increased $V_{max}$ and a pH optimum shift, becoming closer to Mep1 properties. Other specific residues probably enable to accommodate the glutamate in the pore of Mep1 to allow substrate translocation. While Mep2$^{H194E}$ is active, it turns unable to allow filamentation suggesting that the kinetic of substrate translocation, or the pathway followed by the substrate through the pore is crucial to trigger filamentation.

Uncoupling of transport and sensing functions was previously proposed to occur in a Mep2$^{H348A}$-GFP tagged variant with a substitution at position His2 of the histidine-twin, conserving transport activity while losing the ability to allow filamentation [30]. Conservation of ammonium transport activity was also concluded in another study in which the Mep2$^{H348A}$ variant was overexpressed under the control of a strong constitutive promoter [42]. By contrast, an equivalent CaMep2$^{H342A}$-GFP variant was non-functional, but also did not reach the plasma membrane [28]. Our data with non-tagged and not overexpressed Mep2$^{H348A}$ variant are consistent with a reduction of transport activity, precluding to conclude on transport/sensing uncoupling. In ScMep2$^{H194E}$, transport and signaling functions appear uncoupled, but the kinetic of transport and probably the mechanism of transport are modified [29]. Of note, the corresponding CaMep2$^{H188E}$ variant also loses the capability to allow filamentation in *S. cerevisiae* compared to CaMep2, indicating that the signaling mechanism employed by the Mep2 protein from the pathogen is similar.

Our data are consistent with filamentation signaling occurring during the conformational changes of the hydrophobic core of Mep2 accompanying substrate translocation through the pore. This could be sensed by surrounding signaling partners triggering filamentation. It is also conceivable that Mep2 proteins in condition of filamentation induction are transporting a specific substrate, a given molecular entity that must be transported in sufficient amounts to in turn induce the signal leading to filamentation, directly or indirectly. For instance, while characterizing the specific behavior of Mep2$^{H194E}$, we initially proposed that functional subfamilies could co-exist in fungal Mep-Amt-Rh transport proteins, impacting on internal pH in opposed

ways, in turn allowing or not filamentation [29]. The question of the mechanism used by Mep-Amt-Rh to translocate ammonium has been the center of numerous studies and debates [18, 32–37]. Use of N isotope fractionation principles, recently indicated that all Mep-Amt-Rh proteins perform $NH_4^+$ deprotonation during the transport process, followed by $NH_3$ effective translocation [37]. The fate of the released proton remained unsolved. Here, we show that, in exponentially growing cells, Mep1 and Mep2 alter differently the cytosolic pH upon ammonium supplementation, Mep1 tending to acidify whereas Mep2 to alkalinize the cytosol. Ammonium transport by Mep2^{H194E} induces a pronounced acidification, a behavior more similar to Mep1. Moreover, electrophysiology and uptake measurements upon heterologous expression in *Xenopus* oocytes indicate that Mep1 ensures electrogenic transport of ammonium while Mep2 enables electroneutral transport. This could be consistent with Mep1 ensuring the co-translocation of the proton while Mep2 does not. It is tempting to consider the possible existence of fungal Mep proteins showing different substrate transport mechanisms with alternatives regarding proton translocation and with different subsequent impact on filamentation induction. The fate of the proton released by $NH_4^+$ deprotonation could thus vary according to the considered Mep-Amt-Rh protein or even according to the environmental condition for a given protein. Studies using an intramolecular probe sensing conformational changes in Mep2 indicate structural constraints at pH 4 but not at pH 6 upon ammonium addition [54]. We have shown that Mep2 ensures rapid and efficient substrate uptake at both pH in initial uptake rate measurements [11, 29]. Altogether, these data suggest that Mep2 could also be able to translocate substrate by different mechanisms with different conformational outcomes. As mentioned before, two different C-terminally truncated *C. albicans* Mep2 proteins, one of which promotes hyperfilamentation, while the removal of few additional amino acids abolishes filamentation without altering transport efficiency have been described [22]. It is conceivable that the additional truncation has altered the mode of transport from Mep2-like to Mep1-like. The precise physiological and electrochemical context of cells facing the filamentation milieu could favor a given transport mechanism in Mep2, in turn allowing filamentation directly or indirectly. In the tested conditions, with cells growing exponentially in non-limiting conditions, the pH variation associated with substrate transport via Mep proteins was minor. However, small pH changes could nevertheless impact on signaling pathways. pH modifications are known to regulate several cellular processes, such as cell proliferation and differentiation [55–58]. As $pH_c$ homeostasis is sharply controlled, these cell processes are driven by relatively small changes in pH. For example, apoptosis appears to be triggered by pH variations as small as 0.3–0.4 [59]. Moreover, by leaving the proton in the periplasm after $NH_4^+$ deprotonation, Mep2 could contribute to the maintenance of the electrochemical gradient, which could turn advantageous during limitation conditions. By promoting the translocation of the weak and diffusible base $NH_3$, Mep2 transport function could further affect acidic stores of the cells. This could for instance potentially lead to calcium release and subsequent activation of signaling pathways, as proposed in other cellular contexts [9, 60, 61]. There is a growing interest for the role that pH modulation could play in the virulence of pathogenic fungi [62, 63].

In 1998, Lorenz and Heitman proposed, for the first time to our knowledge, that a yeast transporter, Mep2, could play an additional sensor role triggering filamentation signaling [20]. Nowadays, the concept of transceptors, used to define proteins combining a substrate transport capacity and a substrate receptor function coupled to a signal transmission, appears to concern transporters of all kind of micro- and macro- nutrients in eukaryotes, from fungi to human [64, 65]. Albeit an exciting concept, the firm demonstration of the existence of transceptors with separated transport and sensing functions remains highly challenging. Identification of the expected protein partners required to relay the signal to global response systems represents a major task to support the concept. Here, by addressing the transceptor function of

ScMep2, we show that signaling proceeds in the absence of putative partners exclusively binding to the CTD, the largest cytosolic domain of the protein, and that signaling and transport functions are closely intertwined. A distinct electroneutral $NH_3$ transport via ScMep2 implies that the signaling function might be, directly or indirectly, linked to proton-dependent processes pointing to pH as a putative signal mediator.

## Methods

### Strains and growth conditions

The *S. cerevisiae* strains used in this study are listed in S3 Table. All the strains are isogenic with the wild type 1278b [66]. Cells were grown at 29˚C. Cell transformation and gene deletions were performed as described previously [67, 68]. Cells were grown on minimal buffered medium (pH 6.1) supplemented with 3% glucose, vitamins, and trace metals prior to use [69]. Agarose MP (Roche) 1% was used for corresponding solid medium. For cell maintenance, 0.1% glutamate was provided as nitrogen source to avoid selection pressure. For pHluorin assays, 0.1% proline was used as nitrogen source. Pseudohyphal growth tests were performed as previously described [5]. A suspension of diploid cells was patched onto Synthetic Low Ammonium Dextrose (SLAD) and Synthetic High Ammonium Dextrose (SHAD) (0.68% Yeast Nitrogen Base without amino acids and without $(NH_4)_2SO_4$, 3% glucose, 1% agar bacteriological (Oxoid)), respectively supplemented with 50 μM or 0.5 mM $(NH_4)_2SO_4$. For growth tests, diploid cells were streaked on SLAD and SHAD media to follow the formation of colonies. For ammonium removal assays, cells were grown in liquid Synthetic High Proline Dextrose (SHPD) medium (0.68% Yeast Nitrogen Base without amino acids and without $(NH_4)_2SO_4$, 3% glucose, 0.1% proline). The use of the non-preferred source proline has been previously shown to enable high expression of *MEP* genes, and high levels of substrate transport activity when using a minimal buffered medium (pH 6.1) [11, 69]. The *Escherichia coli* JM109 strain was used for plasmid purification and amplification. Bacteria cells were grown at 37˚C in liquid rich medium composed of 0.1% glucose, 1% bactotryptone, 1% bactopeptone, 0.5% yeast extracts, 0.5% NaCl, 0.07% $K_2HPO_4$ and 0.3% $KH_2PO_4$. The corresponding solid medium was supplemented with 1% agarose MP (Roche). To select bacteria successfully transformed by a plasmid, L-ampicilin 50 μg/ml was added to the culture medium.

### Plasmids and mutagenesis

Plasmids used in this study are listed in S4 Table. For electrophysiological analyses, the DNA sequence of Mep1, Mep2 and Mep2-GFP were amplified by PCR using as templates YCp-Mep1, YCpMep2 and p416 Gal1-Mep2-GFP plasmids, respectively. All the plasmids were restricted with BamHI and SpeI enzymes and the recovered inserts were ligated into pOO2, a pBF-derived oocyte expression plasmid. The pMep2-pHluorin plasmid was constructed in two steps. First, GFP was replaced by the pHluorin using *in vivo* recombination in the p416 Gal1--Mep2-GFP plasmid linearized with HpaI. In the second step, the *GAL1* promoter of the obtained plasmid was replaced by the native *MEP2* promoter (-660 to -1) by restriction-ligation steps in the SacI and SpeI restriction sites. Site-directed mutageneses of *MEP2* were performed using the QuikChange II Site-Directed Mutagenesis Kit (Agilent Technologies), as described by the provider. Plasmid extraction from bacterial cells was performed using the GeneJET Plasmid Miniprep Kit (Thermo Fischer). CaMep2 (orf19.5672), containing the Sc*MEP2* promoter (-400 to -1) and terminator (1 to 262), and the mutated CaMep2[H188E] genes were synthesized and cloned in pFL38 and pFL44 vectors by GeneCust.

All constructs were verified by sequencing. The sequence of all the primers used in this study are available upon request.

## Ammonium removal assay

Cells were grown to exponential phase in SHPD medium. At time 0, 250 μM $(NH_4)_2SO_4$ was added to the culture and samples of culture supernatant were withdrawn at time intervals and assayed for ammonium concentration by enzymatic coupling with L-glutamate dehydrogenase (Roche) [14]. The experiment was started at the same $OD_{660}$ for all cultures.

## Western Immunoblotting

Membrane-enriched cell extracts were prepared as previously described [39]. For PNGase F treatment on membrane-enriched cell extracts, the collected membrane pellet was suspended in buffer (1X PBS, 10 mM EDTA pH8, 0.5% octyl glucopyranoside, 0.2% 2-mercaptoethanol, 3 mM PMSF and proteinase inhibitors) and incubated 1 h at 37°C in the presence of 1.5 unit of peptide-N-glycosidase F (PNGase F, Sigma-Aldrich, Roche). Proteins were precipitated with 10% TCA. For blot analysis, equal protein amounts (≈20 μg) were loaded onto a 7% SDS-polyacrylamide gel in a Tricine system [70]. After transfer to a nitrocellulose membrane, Mep2 proteins were probed with a rabbit antiserum (1:1000) raised against the C-terminal region of Mep2 [19]. Pma1 proteins were probed as loading control with anti-Pma1 antibodies (1:10000) [71]. Primary antibodies were detected with horseradish peroxidase-conjugated anti-rabbit-IgG secondary antibodies followed by measurement of chemoluminescence (Lumi-Light$^{PLUS}$, Roche).

## *In vivo* pH measurement

Cells were transformed with pYES-pHluorin or pMep2-pHluorin plasmids. For *in situ* pHluorin calibration, about $10^7$ cells, exponentially growing in buffered minimal proline medium, were centrifuged for 5 minutes at 14,000 rpm and resuspended in PBS containing 0.1 mg/ml digitonin. After 15 minutes of gentle agitation at room temperature, cells were centrifuged 5 minutes at 14,000 rpm and resuspended in citric acid/$Na_2HPO_4$ buffer of pH values ranging from 5.2 to 8.2. pHluorin fluorescence emission at 510 nm was measured at two excitation wavelengths (395 and 475 nm) using an Infinite M200pro (Tecan) multimode microplate reader. The ratio of emission intensity resulting from excitation at 395 and 475 nm was calculated, plotted against the corresponding buffer pH and fitted to a third-degree polynomial regression curve. For *in vivo* pH measurements, cells expressing pHluorin were grown in buffered minimal proline medium to the exponential phase and 200 μl of culture containing about $0.2x10^7$ cells were used. Fluorescence emitted at 510 nm was directly measured at two excitation wavelengths (395 and 475 nm), and the ratio of fluorescence intensity values (F395/F475) was calculated. The pH values were calculated by extrapolating the ratio values on the calibration curve. In ammonium treatment experiments, 20 μl of medium containing ammonium at a concentration enabling to reach the desired final concentration was added to the proline-grown cells. Cells treated with ammonium-free medium were used as negative control. Evolution of pH was corrected for negative control variation (ΔpH) by subtracting for each time point the pH values of the negative control from pH values of interest conditions. All pH determination experiments were repeated at least three times. The standard deviations are indicated for the values corresponding to the averages of the independent experiments.

## Electrophysiological measurements and ammonium uptake in Oocytes

The electrophysiological methods are described in more detail elsewhere [72]. Briefly, oocytes were ordered at Ecocyte Bioscience (Castrop-Rauxel), presorted again and injected with 50 nl of cRNA (0.8 μg/μl). Oocytes were kept in ND96 for 4 days at 18°C and then placed in a small recording chamber. The recording solution was 110 mM choline chloride, 2 mM $CaCl_2$, 2 mM

MgCl$_2$, and 5 mM MES, pH adjusted to 5.5 with Tris. Variable ammonium concentrations were added as NH$_4$Cl salt. Currents without added ammonium were subtracted at each voltage. For determination of ammonium uptake, the oocytes were incubated for 30 min in recording solution additionally containing 3 mM 98% $^{15}$N added as ammonium sulfate salt. The oocytes were washed six times in H$_2$O before they were separately put in tin cups for isotope ratio mass spectrometry measurements. Cups were balanced before usage and after freeze-drying of the oocyte. The cup with oocyte was closed, balanced again and used for $^{15}$N and total N determination by isotope ratio mass spectrometry.

## Photomicroscopy

Pictures of yeast colonies were taken directly from Petri plates using a Zeiss Axio Observer Z1 microscope, driven by MetaMorph (MDS Analytical Technologies), with a 10x primary objective and a 2.5x camera adaptor.

## Fluorescence microscopy

Images of yeast cells were acquired using a Zeiss LSM710 laser-scanning confocal microscope, equipped with the Airy scan module. Acquisitions were performed using the ZEN 2.1 software and images were processed using ImageJ. For localization analysis of Mep-GFP fusion constructs in oocytes, the latter were kept in ND96 and then directly analyzed under an LSM700_ZEN_2010 microscope (Zeiss). The excitation wavelength was 488 nm; emission was detected for GFP between 500 and 550 nm.

## Structure and sequence analyses

The images of the ScMep2 structure [15] (PDB id 5AEX, chain A) were produced using a combination of MolScript and Raster3D programs. The non-covalent interactions were defined with the ARPEGGIO program.

The protein stability changes ($\Delta\Delta$G) upon single-point mutation were predicted using 7 different programs: INPS3D, mCHS, I-mutant3.0, ENCom, SDM, DUET and Dynat-Mut.

For the WebLogo analyses, two datasets were retrieved from the UniRef100 database by Blast searches with the ScMep1 (sp:P40260) and ScMep2 (sp:P41948) as searching sequences, respectively. Multiple sequence alignments were performed with the ClustalW2 program. Sequences containing too many gaps in the multiple alignments were removed. Sequences were subdivided according to secondary structures and subdomain limits. The WebLogo server was then used to generate the different plots.

## Supporting information

**S1 Fig. Reduced transport function of poorly active Mep2 CTD variants correlates with reduced pseudohyphal growth efficiency.** (A-B) Homozygous diploid *mep2Δ* (ZAB2) cells were transformed with the pFL38 empty plasmid (-) or with YCpMep2, YCpMep2$^{S426stop}$, YCpMep2$^{CΔ428−431}$, YCpMep2$^{CΔ428−449}$, YCpMep2$^{CΔ434−449}$, YCpMep2$^{CΔ434−485}$, YCpMep2$^{H199Y, S426stop}$, YCpMep2$^{H194E}$ and YCpMep2$^{H348A}$. (A) Growth tests on SLAD and SHAD media at day 3 (3d) and 7 (7d) at 29˚C. (B) Pseudohyphal growth tests on SLAD and SHAD media at day 7 at 29˚C.
(TIF)

**S2 Fig. WebLogo plots of the structural segments in Mep1-like vs Mep2-like sequences.** Two datasets were retrieved from the UniRef100 database by Blast searches with the ScMep1 (sp:P40260) and ScMep2 (sp:P41948) as searching sequences, respectively. No sequence was in

common to the two datasets by considering the 350 top hits reported. Multiple sequence alignments were performed with the ClustalW2 program. Sequences containing too many gaps in the multiple alignments were removed, resulting in 348 sequences for both pools of sequences. Sequences were subdivided according secondary structures and subdomain limits, giving the following structural segments, from N- to C-terminal, the external N-terminus, the transmembrane helix #1 (TM1), the cytoplasmic loop #1 (CL1), TM2, the external loop #2 (EL2), TM3, CL3, TM4, EL4, TM5, CL5, TM6, EL6, TM7, CL7, TM8, EL8, TM9, CL9, TM10, EL10, TM11, the cytoplasmic C-terminal proximal linker, the cytoplasmic enhancer subdomain, the cytoplasmic C-terminal linker subdomain, the cytoplasmic autoinhibitory subdomain and the distal part of the cytoplasmic C-terminus. Note that the CTD of ScMep1 was subdivided according sequence comparison with ScMep2. The WebLogo server (https://weblogo.berkeley. edu) was then used to generate the different plots, one for each structural segment. The logo consists of amino acid stack for each position in the aligned sequence dataset. The height of the stack indicates the sequence conservation at that position (maximum = 4 bits), while the height of the amino acid one-letter symbol within the stack shows the relative frequency of the amino acid at that position. The color scheme is aromatic amino acids (F, H, W, Y) in green, polar (N, Q, S, T) in purple, negatively charged (D, E) in red, positively charged (K, R) in blue, aliphatic (A, I, L, M, V) in black, cysteine (C) in brown, proline (P) in grey and glycine (G) in orange. Sequence numbering is according ScMep1 or ScMep2 sequences.
(PDF)

**S1 Table. Recapitulative table of the transport and signaling functions of the different Mep2 variants used in this study.** The localization of mutations in the Mep2 topology is indicated (CTD, cytoplasmic C-terminal domain; TM, transmembrane domain; EL, extracellular loop). The growth on SLAD and SHAD was determined from the growth tests shown in Figs 2A, 3A, 4B, 9A and S1A. The ammonium removal capacity was determined from Figs 1C and 4C–4E. The pseudohyphal capacity was determined from Figs 2B, 3B, 3E, 4A, 9B and S1B. Symbols for growth: ++, strong; +, intermediate or like wild-type; +/-, weak; -, absence of growth. Symbols for ammonium transport: ++, very strong; +, intermediate or like wild-type; +/-, low; -, very low. Symbols for pseudohyphal growth: ++, very strong; +, visible; +/-, weak; -, absence. ND, not determined.
(PDF)

**S2 Table. Predictions of Mep2 protein stability change upon four point mutations considered in the current paper.** The Protein stability changes (ΔΔG) were predicted using 7 different programs. The ΔΔG values are indicated in kcal/mol and were categorized into "Highly destabilizing" (ΔΔG ≤ -2 kcal/mol), "Destabilizing" (-2 kcal/mol < ΔΔG < -0.5 kcal/mol), "Neutral" (-0.5 kcal/mol ≤ ΔΔG ≤ +0.5 kcal/mol), "Stabilizing" (+0.5 kcal/mol < ΔΔG < +2 kcal/mol) and "Highly stabilizing" (ΔΔG ≥ +2 kcal/mol). A prediction consensus was generated by averaging the 7 results. The ΔΔG values for the consensus were indicated as mean ± SEM, where SEM is the standard error of the mean.
(PDF)

**S3 Table. The *S. cerevisiae* strains used in this study.**
(PDF)

**S4 Table. The plasmids used in this study.**
(PDF)

## Acknowledgments

We thank Pascale Van Vooren, Silvia Soto Diaz, Sandra Lecomte, Gilles Vanderstocken and Patrice Godard for support and discussions. We thank Johan Thevelein for sharing Mep2

expression plasmids. Further, we thank Deborah Schnell for technical assistance in the oocyte experiments as well as Uwe Ludewig for support and discussions.

## Author Contributions

**Conceptualization:** Ana Sofia Brito, Anna Maria Marini, Mélanie Boeckstaens.

**Data curation:** Ana Sofia Brito.

**Formal analysis:** Ana Sofia Brito, Benjamin Neuhäuser, René Wintjens, Mélanie Boeckstaens.

**Funding acquisition:** Anna Maria Marini.

**Investigation:** Ana Sofia Brito, Benjamin Neuhäuser, René Wintjens, Mélanie Boeckstaens.

**Methodology:** Ana Sofia Brito, Mélanie Boeckstaens.

**Project administration:** Anna Maria Marini, Mélanie Boeckstaens.

**Resources:** Benjamin Neuhäuser, Anna Maria Marini.

**Supervision:** Anna Maria Marini, Mélanie Boeckstaens.

**Visualization:** Ana Sofia Brito, Benjamin Neuhäuser, René Wintjens, Mélanie Boeckstaens.

**Writing – original draft:** Ana Sofia Brito, Benjamin Neuhäuser, René Wintjens, Anna Maria Marini, Mélanie Boeckstaens.

**Writing – review & editing:** Ana Sofia Brito, Benjamin Neuhäuser, René Wintjens, Anna Maria Marini, Mélanie Boeckstaens.

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
