## [Decision Letter · Decision Letter 0]

27 Jun 2019

Dear Anna Maria,

Thank you very much for submitting your interesting Research Article entitled 'Yeast filamentation signaling is connected to a specific substrate translocation mechanism of the Mep2 transceptor' to PLOS Genetics.  Your manuscript was fully evaluated at the editorial level and by independent peer reviewers who are all experts in the field.  The reviewers appreciated the attention to an important problem, but raised some concerns about the current manuscript.  Based on the reviews, we will not be able to accept this version of the manuscript, but we would be willing to review again a revised version.  We cannot, of course, promise publication at that time.

The reviewers found your findings and the model proposed interesting and thought provoking.  This said, they have outlined ways that the model could be tested further experimentally that could provide additional experimental support for the model for signaling advanced.  There are also areas in which they suggest controls, or ways in which expression levels may alter the interpretations suggested for certain mutants under certain experimental conditions.  In aggregate, the summary of the reviewers' suggestions support reconsideration after revision, and in some ways these might be considered major revisions given that additional experiments are suggested and may be required to address points raised.  I have adjusted the time suggested for response accordingly below (to 90 to 120 days) to provide sufficient time to address the reviews as fully as possible.  A revised manuscript would be sent for re-review, and ideally with the same set of reviewers given availability.

Should you decide to revise the manuscript for further consideration here, and we very much hope you do as we are quite interested in your studies and the model presented, your revisions should address the specific points made by each reviewer. We will also require a detailed list of your responses to the review comments and a description of the changes you have made in the manuscript.

If you decide to revise the manuscript for further consideration at PLOS Genetics, please aim to resubmit within the next 90 to 120 days, unless it will take extra time to address the concerns of the reviewers, in which case we would appreciate an expected resubmission date by email to plosgenetics@plos.org.

[LINK]

We are sorry that we cannot be more positive about your manuscript at this stage, and we hope that you find the reviews helpful and constructive. Please do not hesitate to contact us if you have any concerns or questions.

Yours sincerely,

Joe

Joseph Heitman, MD, PhD (heitm001@duke.edu)

Associate Editor

PLOS Genetics

Gregory P. Copenhaver

Editor-in-Chief

PLOS Genetics

Reviewer's Responses to Questions

**Comments to the Authors:**

Reviewer #1: In this thought-provoking manuscript, Brito, et al., examine the transport and signaling mechanisms of the Mep2 ammonium transceptor in yeast. They show that transport is required for signaling and that the cytoplasmic C-terminal domain, previously thought to be required for the signaling function, is dispensable in certain circumstances. The mutational studies are extensive, based on previously characterized sequences combined with some new structural analysis. Interestingly, the genetic experiments are coupled with measures of transport and cytoplasmic pH (in yeast) and electrochemical current (in Xenopus). These support a difference in transport mechanism between the signaling Mep2 and non-signaling Mep1 proteins, concluding that Mep1 transports the ammonia molecule with its proton and Mep2 beings in only the proton.

These are intriguing conclusions based on solid and rigorous experiments. They do not identify the elusive signaling mechanism, but the paper does lead to new directions for understanding it. The electrophysiology in Xenopus is particularly persuasive. The subject matter and approach are appropriate for PLoS Genetics. A few, mostly minor, comments are below.

1. The weakest experimental aspect of the paper is the measurements of cytosolic pH. There are believable differences between Mep1 and Mep2, but they are quite small. This is likely because of the proton pumps in the plasma and vacuolar membranes. Perhaps if these were inhibited, pharmacologically or genetically, the differences would be enhanced and the confounding pH recovery phase reduced.

2. The genetic experiments are thorough, but it is a bit confusing to keep track of all the mutants and what they are. A cartoon of the linear protein sequence, with the TMs, the enhancer, AI domains, and the position of the point mutants would be very helpful. Perhaps this could be added at the top of Fig. 1?

3. Are they sure that the Mep2 enhancer domain mutants, which have reduced transport and signaling activity are stably expressed? A Western would be confirmatory (was this not done because the C-terminal antibody does not recognize these mutants well?).

4. Some of the figures (especially Fig. 1) have many constructs tested and it is challenging to really sort out which line represents which mutant . A consistent color scheme would help greatly. Alternatively, splitting this into two figures with the enhancer domain mutants in one and the others in another might help.

Reviewer #2: Filamentation is the critical process for fungal development and pathogenicity of many fungal pathogens. In many fungi, filamentation is triggered by nitrogen starvation and therefore Mep2, ammonium permease, has been implicated in the process. A question remains whether Mep2 has a ‘transceptor’ activity for the filamentation process. That is to say, does Mep2 not only sense ammonium, but also transport it for filamentation? In this paper, Brito et al. tackled this question through genetic, molecular, and structure analyses of yeast Mep2. Through domain deletion experiments of Mep2 in triple mep deletion mutant background, the authors clearly demonstrated that ammonium transport function is tightly correlated with filamentation efficiency. They showed that the phosphomimetic mutation S457D in the C-terminal autoinhibitory domain of Mep2 did not lock Mep2 as a constitutive filamentation active form and it still required substrate translocation. Previously, it has been shown that the C-terminal domain (CTD) plays a role in tuning the transport activity of Mep2. Here the authors demonstrated that the CTD domain is not absolutely required for filamentation induction, because the H199Y and G349C suppressor mutation in the transmembrane domains can trigger filamentation even without the CTD domain. Based on the analysis based on reported X-ray structure of ScMep2, H199 and G349, which are evolutionarily divergent and conserved among other fungal orthologs, respectively, are facing each other. Subsequently, they observed the different impact of Mep1 and Mep2 on cytosolic pH. Supporting this, electrophysiology analyses using heterologous expression of Mep1 and Mep2 in Xenopus oocytes demonstrated that non-filamentation signaling Mep1 mediated electrogenic transport while Mep2 mediated electroneutral substrate translocation.

Overall, this study provides very informative and valuable data for confirming the transceptor activity of Mep2 required for yeast filamentation. Here the authors clearly demonstrated that the Mep2 transceptor not only functions as ammonium transporter and a signaling receptor for filamentation, even without involvement of cytosolic partner proteins. So intracellular pH change can be a signal mediator for yeast filamentation. The manuscript is very well written and most of conclusions are nicely supported by a sophisticated set of experiments. I have the following minor comments.

[Minor comments]

Each paragraph should be indented throughout the manuscript.

The first paragraph of the Introduction section is too long. It needs to be divided into two paragraphs (line 65?)

Line 76-78: The meaning of this sentence is unclear. Please clarify it with more detailed explanation.

Lines 132-141: A graphic illustration of each domain in the Mep2 protein (in Figure 1?) should make readers better understood. In addition, any previously known model describing how Mep2 protein works in association with Npr1 kinase could help readers too.

It seems that only two independent experiments were performed for Figure 1B and Figure 3C experiments. For statistical analysis, at least three biological independent experiments should be performed and proper statistical analysis should be done.

The graphs in Figure 2B are very difficult to see. Please use colored lines with more distinct symbols.

Each Figure 5 graph should be separately plotted at 2 mM and 20 mM NH3, as the two data are overlapped and hard to discriminate.

Reviewer #3: This study aimed at elucidating which specific aspects of the ammonium permease Mep2, as opposed to its paralogs Mep1 and Mep3, are responsible for its ability to induce filamentous growth in response to nitrogen limitation in the budding yeast Saccharomyces cerevisiae. By testing the ability of mutated forms of Mep2 to mediate ammonium transport and promote filamentation, the authors confirmed their previous observation that the induction of filamentous growth by Mep2 generally correlates with its ammonium transport activity. The C-terminal cytoplasmic tail of Mep2 was found to be dispensable for the induction of filamentous growth by some mutated forms of the protein, and the authors conclude that it does not represent a signaling domain. Electrophysiological studies showed that ammonium transport by Mep2 is electroneutral (i.e., it transports NH3 without H+ after deprotonation of NH4+) whereas transport by Mep1 is electrogenic (i.e., the proton is co-transported). Accordingly, the authors observed a slight increase in the cytosolic pH after ammonium addition to cells containing Mep2, which was weaker in cells producing only Mep1. The authors conclude that pH modification is the mechanism by which Mep2 induces filamentous growth.

This is a very interesting study that addresses an unresolved question, how Mep2 and related ammonium permeases in other fungi induce morphogenesis.There are a number of issues that in my opinion affect the authors’ main conclusions and should be considered.

Comments

1) It is difficult to envisage that the very minor (less than 0.05 pH units) and presumably short-term (minutes) increase in the cytosolic pH after the addition of ammonium to the cells is the trigger for filamentous growth, considering that the induction of filamentous growth occurs after several days of growth under limiting nitrogen conditions; one would assume that the cells maintain intracellular pH homeostasis under these conditions. Furthermore, the experiments shown in Fig. 5 were performed in the presence of 2 mM ammonium, a concentration that suppressed filamentous growth in the present study (see Fig. 1C, Fig. 2B and E, Fig. 3B). At the low concentrations of ammonium that induce filamentation, one would expect a still smaller, if any, effect on cytosolic pH by Mep2-mediated ammonium uptake. Of note, Mep2 also induces filamentous growth at low concentrations of other nitrogen sources, and this is believed to be due to Mep2-mediated retrieval of excreted ammonium. Under these conditions, even less extracellular ammonium will be present that can be taken up by Mep2 and affect the cytosolic pH. Although the authors also state that conformational changes in Mep2 that occur during ammonium transport are involved in signaling, the sentences at the end of the abstract, author summary, introduction, and discussion convey the message that Mep2-mediated pH changes are the key signal that induces filamentation.

2) The presentation of the results shown in Figs. 1 to 3 could be considerably improved. For one thing, the growth differences between the various strains are not evident in Fig. 1C, Fig. 2B and E, and Fig. 3B, although the triple mepΔ mutants should not grow on SLAD medium. I assume that the triple mepΔ mutants and some of their derivatives formed tiny colonies, and the pictures were taken at different magnifications to show the colony borders, although this is nowhere stated in the manuscript. Growth differences are illustrated in Fig. 1A, Fig. 2A and D, and Fig. 3A, but these pictures appear not very reliable, as is evident from the fact that the mep2Δ single mutant seems to have a growth defect on SLAD medium that is rescued after reintroduction of wild-type MEP2 (a growth defect of the mep2Δ single mutant is unexpected, because previous studies reported that Mep1 alone is sufficient to support wild-type growth). I suggest the following modifications to these experiments and their presentation:

a) Growth and filamentation of the various strains on solid media should be simultaneously illustrated by showing representative single colonies of all strains at equal magnification. I notice that the filamentation phenotype of the wild type on SLAD medium was generally weak in the authors’ experiments, with only occasional pseudohyphae at the colony borders, whereas in other studies, S. cerevisiae colonies grown on SLAD plates are completely surrounded by a filamentous fringe (see, for example, the original study by Lorenz and Heitman describing the role of Mep2 in filamentous growth).

b) The effect on filamentation by all Mep2 variants should also be tested in a mep2Δ single mutant background (as has been done for some mutants in Fig. 2B) to exclude an influence of reduced growth on filamentation.

c) The doubling times of the strains in liquid SLAD medium should be determined to allow a better quantification of the effect of the different Mep2 mutations on growth for comparison with the effect on ammonium transport, which was also determined for cells growing exponentially in liquid medium.

d) Finally, I found it very difficult to find the data for some strains in Fig. 1B, because the symbols cannot be easily distinguished. This could be amended by using colored symbols.

3) I think that the authors should tone down their conclusion in the abstract that the C-terminal tail of Mep2 is not required for the induction of filamentation. Despite the fact that the additional H199Y mutation in the truncated Mep2S426stop protein restored ammonium uptake to wild-type levels (Fig. 3C), filamentation was extremely poor (Fig. 3B). Therefore, the function of the C-terminal tail is not only to upregulate substrate translocation, but it is also required for normal signaling at wild-type transport levels. Signaling may require interactions between the C-terminal tail and other cytoplasmic domains of Mep2 (this has been suggested for C. albicans Mep2), and conformational changes caused by the H199Y mutation may have allowed some signaling to occur in the absence of these interactions (or allowed such interactions to occur with the rest of the C-terminal tail that remains in the truncated protein).

4) The H194E mutation is highly interesting in that it abolishes filamentation, although it increases ammonium transport, a striking exception to the generally observed correlation between ammonium transport and signaling capacity by Mep2. The authors argue that the H194E mutation altered the mechanism of ammonium transport by Mep2 into a Mep1-like mechanism, as already suggested in their previous study. Unfortunately, this mutated protein could not be successfully produced in oocytes, which would have allowed to test if the mutation changed ammonium transport from electroneutral to electrogenic (as a minor note, the sentence in lines 397-399 should be changed, because the H194E variant was not tested in the experiments shown in Fig. 6E). As this is a critical aspect of the present study, I think that the authors should at least test if Mep2 with the H194E mutation shows a Mep1-like behavior instead of a Mep2-like behavior with regard to cytosolic alkalinization (see Fig. 5). If this test can reliably distinguish between the two transport mechanisms, the authors could prove or refute this hypothesis.

5) Although the study focuses on Mep2 of S. cerevisiae, the paper implies that the conclusions about Mep2 function derived from this study extend to its homologs in other fungi. Nevertheless, the authors point out striking differences between the Mep2 proteins of S. cerevisiae and C. albicans, which may hint to different signaling mechanisms by the two proteins. In C. albicans, the C-terminal cytoplasmic tail of Mep2 is required for the induction of filamentous growth even when ammonium transport by mutant proteins appears normal. Mep2 proteins that were C-terminally truncated after amino acid 440 and 423 allowed wild-type growth and exhibited comparable ammonium transport activity. However, while the former promoted hyperfilamentation (caused by the absence of an autoregulatory domain, increased expression levels, or both), the latter was unable to induce filamentation. According to the model proposed in the present study, the removal of the 17 additional amino acids should have altered the mechanism of ammonium transport (to a Mep1-like mechanism) to explain the loss of signaling. The C. albicans mutants are available and could be tested and compared in the same way as the S. cerevisiae mutants. Since the authors repeatedly highlight the relevance of the transition from yeast to filamentous growth for the virulence of pathogenic fungi (and our understanding of its regulation), it seems worthwile to perform and include these experiments.

Reviewer #4: In this manuscript the authors build on their previous studies of the Mep2 transceptor from Saccharomyces cerevisiae. Three main findings are presented: that signalling is linked to transport activity, that the Mep2 CTD is not required for signalling and that signalling involves pH changes within the cytosol. In support of the latter, the authors present evidence that the Mep2 transceptor and the non-signalling Mep1 transporter induce different cytosolic pH profiles. Furthermore, the authors establish that transport by Mep2 is electroneutral whereas transport by Mep1 is electrogenic. Together these data support the model that Mep2 imports ammonia gas that raises cytosolic pH. This is an important finding. What is disappointing is that the authors have not gone one step further and tested some predictions of this model. For example, if the Mep2 dependent pH change regulates signalling then Mep2 variants that transport but do not signal should generate a different pH profile. This may be because transport by the variant is electrogenic or if it remains electroneutral the amount of imported ammonia differs from that imported by wild-type Mep2. Too little or too much ammonia may result in a pH change that does not support signalling. The authors could test the effect on cytosolic pH of the Mep2H194E variant as they have determined that transport by it is electroneutral. Similarly, the hyperactive Mep2G349C variant would be expected to generate a different pH profile from that of the Mep2H194E variant. Other points of discussion could include those Mep2 homologues that act as signalling proteins in S. cerevisiae, such as Ump2 from U. maydis. These should have the same impact as Mep2 on cytosolic pH via electroneutral transport when expressed in S. cerevisiae.

Minor comments

lines 131-198

The fact that certain Mep2 variants are more active in the absence of Npr1 may be due to their increased expression and could be tested. A previous study by the authors (Boeckstaens et al., 2013) established that the Mep2 levels are higher in the absence of Npr1.

lines 191-131

The authors conclude that the Mep2S457D variant does not lock Mep2 into a constitutively active conformation as it does not induce filamentation when cells are grown on SHAD plates. Expression of this variant would be expected to be reduced under SHAD conditions. Wild-type Mep2 needs to be highly expressed if it is to signal in response to low ammonium. A variant locked in a constitutively active conformation may therefore loose its ability to signal if expressed at a low level. The relative levels of Mep2S457D in response to SLAD and SHAD should be determined.

**Have all data underlying the figures and results presented in the manuscript been provided?**

Reviewer #1: Yes

Reviewer #2: Yes

Reviewer #3: Yes

Reviewer #4: Yes

PLOS authors have the option to publish the peer review history of their article (what does this mean?). If published, this will include your full peer review and any attached files.

Reviewer #1: No

Reviewer #2: No

Reviewer #3: No

Reviewer #4: No

---

## [Decision Letter · Decision Letter 1]

18 Dec 2019

Dear Anna Maria,

Thank you very much for submitting your revised Research Article entitled 'Yeast filamentation signaling is connected to a specific substrate translocation mechanism of the Mep2 transceptor' to PLOS Genetics. Your manuscript was fully evaluated at the editorial level and by the same four independent peer reviewers who reviewed the original submission. The reviewers appreciated the attention to an important topic but identified some aspects of the manuscript that should be improved.  See further comments from the associate editor below.

We therefore ask you to modify the manuscript according to the review recommendations before we can consider your manuscript for acceptance. Your revisions should address the specific points made by each reviewer.

[LINK]

Yours sincerely,

Joe

Joseph Heitman, MD, PhD

Associate Editor

PLOS Genetics

Gregory P. Copenhaver

Editor-in-Chief

PLOS Genetics

Associate Editor comments:

The revised manuscript has been reviewed by the four original reviewers. All of the reviewers found the manuscript responsive to reviews, and improved. Three recommended acceptance. Reviewer 3 raised a few remaining issues that we request you address by revision, and this constitutes now minor compared to major revision. We appreciate your efforts to respond to the reviews, and improve the manuscript.

Reviewer's Responses to Questions

**Comments to the Authors:**

Reviewer #1: I am impressed by the extensive and thoughtful changes to the manuscript in response to the previous suggestions, particularly the clever pHluorin fusion experiments. I have no further reservations regarding the manuscript, though the authors may want to re-edit the paragraph at the bottom of page 17, as there are several typos.

Reviewer #2: The authors nicely addressed all of my comments in this revised manuscript. I do not have any more comments. Nice work!

Reviewer #3: In their revised manuscript, the authors went to great lengths to address the reviewers’ comments. Therefore, I would not request additional experiments, but there are several issues that in my opinion should be clarified.

1) I still have difficulties to see how the short-term changes in intracellular pH that were observed upon addition of ammonium to proline-grown cells can be responsible for the induction of filamentous growth after several days under nitrogen-limiting conditions, as the cells should re-establish and maintain intracellular pH homeostasis. The authors may also explain why there is an initial drop in intracellular pH upon ammonium uptake, the reason for which is not clear to me. Although Mep1 transports the proton together with ammonia, one would expect that ammonia is immediately protonated again within the cell, so that there is no net increase in the intracellular concentration of protons by Mep1-mediated ammonium transport. In the case of Mep2, ammonia transport without the dissociated proton should cause an initial intracellular alkalinisation, because cytosolic protons are captured by the internalized ammonia. Which mechanisms are responsible for the actually observed pH changes?

2) The authors point out that some Mep2 CTD variants showed an even better complementation efficiency in the absence of Npr1 than in its presence, and that this correlated with an improved ammonium removal rate in SHPD and also with an increased capacity to induce filamentation on SLAD (Page 9). This appears to be the case for the variants shown in the upper panels of Fig. 1C. However, the Δ442-449, Δ442-485, and Δ450-485 variants, which all confer a strong hyperfilamentous phenotype in the absence of Npr1 (Fig. 2B) do not show better ammonium uptake in the absence of Npr1 than in its presence (Fig. 1C, lower panels). If anything, the uptake rate is slightly lower in the absence of Npr1 (this is now better visible in the revised figure). Therefore, stronger filamentation does not correlate with more efficient ammonium uptake in these cases.

3) In a similar vein, the repeatedly emphasized correlation between transport and filamentation efficiencies does also not hold in several other cases. The H199Y S426stop mutation did not impair transport (it was even slightly increased, Fig. 4C), but almost completely abolished filamentation (Fig. 4A, top panels). Furthermore, the G349C mutation strongly increased transport (Fig. 4D) without resulting in enhanced filamentation in NPR1 wild-type cells (Fig. 4A, top panels). This should be more critically discussed.

4) In my previous comments I had suggested that the authors compare the mode of ammonium transport by two different C-terminally truncated C. albicans Mep2 proteins, one of which promotes strong hyperfilamentation, while the removal of few additional amino acids abolished filamentation without altering transport efficiency. According to the model proposed in the present study, the additional truncation should have altered the mode of transport from Mep2-like to Mep1-like. Instead, the authors compared wild-type CaMep2 and a mutated form, in which H1 of the conserved histidine-twin in the conducting pore was changed into glutamate found in Mep1-like proteins, upon heterologous expression in S. cerevisiae. The finding that the mutation abolished filamentation is taken as evidence that a conserved transport mechanism explains the ability of both ScMep2 ans CaMep2 to promote filamentous growth. However, the authors disregard (see line 451) that the mutated Mep2 restored growth of SCmepΔ triple mutants on SLAD plates much less efficiently than did wild-type CaMep2 (Fig. 9A). It is likely that the mutated protein was less stable than the wild-type protein, explaining its inability to promote filamentation.

Reviewer #4: -

Associate editor note, this reviewer did not have detailed comments for the authors and found the manuscript improved and suitable for acceptance.

**Have all data underlying the figures and results presented in the manuscript been provided?**

Reviewer #1: Yes

Reviewer #2: Yes

Reviewer #3: Yes

Reviewer #4: Yes

PLOS authors have the option to publish the peer review history of their article (what does this mean?). If published, this will include your full peer review and any attached files.

Reviewer #1: No

Reviewer #2: No

Reviewer #3: No

Reviewer #4: No

---

## [Editor Report · Decision Letter 2]

28 Jan 2020

Dear Anna Maria,

We are pleased to inform you that your re-revised manuscript entitled "Yeast filamentation signaling is connected to a specific substrate translocation mechanism of the Mep2 transceptor" has been editorially accepted for publication in PLOS Genetics. Congratulations! We appreciate the care and attention to detail throughout the revision process.

Yours sincerely,

Joseph Heitman, MD, PhD

Associate Editor

PLOS Genetics

Gregory P. Copenhaver

Editor-in-Chief

PLOS Genetics

Comments from the reviewers (if applicable):

**Data Deposition**

http://datadryad.org/submit?journalID=pgenetics&manu=PGENETICS-D-19-00855R2

**Press Queries**

---

## [Editor Report · Acceptance letter]

12 Feb 2020

PGENETICS-D-19-00855R2 

Yeast filamentation signaling is connected to a specific substrate translocation mechanism of the Mep2 transceptor 

Dear Dr Marini, 

We are pleased to inform you that your manuscript entitled "Yeast filamentation signaling is connected to a specific substrate translocation mechanism of the Mep2 transceptor" has been formally accepted for publication in PLOS Genetics! Your manuscript is now with our production department and you will be notified of the publication date in due course.

With kind regards,

Matt Lyles

PLOS Genetics

On behalf of:
